# Generation of cryopreserved macrophages from normal and genetically engineered human pluripotent stem cells for disease modelling

**Christie Munn**[1], **Sarah Burton**[1☯], **Sarah Dickerson**[1☯], **Kiranmayee Bakshy**[1],
**Anne Strouse**[2], **Deepika Rajesh**[1]*

**1** FUJIFILM Cellular Dynamics, Inc., Madison, WI, United States of America, **2** PPD® Laboratories, Middleton, WI, United States of America

☯ These authors contributed equally to this work.
* deepika.rajesh@fujifilm.com

**Data Availability Statement:** All relevant data are presented within the manuscript and its Supporting Information files. The GEO link accompanied with

## Abstract

Macrophages are innate immune cells that play critical roles in tissue homeostasis, inflammation, and immune oncology. Macrophages differentiated from human induced pluripotent stem cells (iPSCs) overcome many limitations of using peripheral blood derived macrophages. The ability to scale up and cryopreserve a large amount of end stage macrophages from single clonal iPSCs from normal and disease specific donors offers a unique opportunity for genomic analysis and drug screening. The present study describes the step wise generation and characterization of macrophages from iPSCs using a defined serum free method amenable to scale up to generate a large batch of pure end stage cryopreservable macrophages expressing CD68, CD33, CD11c, CD11b, CD1a, HLA-DR, CD86, CD64, CD80, CD206, CD169, CD47, HLA-ABC, and CX3CR. The end stage macrophages pre and post cryopreservation retain purity, morphology, responsiveness to stimuli and display robust phagocytic function coming right out of cryopreservation. The same differentiation process was used to generate end stage macrophages from isogenic iPSCs engineered to mimic mutations associated with Parkinson's disease (SNCA A53T), neuronal ceroid lipofuscinosis (GRN2/GRN R493X), and Rett syndrome (MECP2-Knockout). End stage macrophages from isogenic engineered clones displayed differential macrophage-specific purity markers, phagocytic function, and response to specific stimuli. Thus, generating a panel of functional, physiologically relevant iPSC-derived macrophages can potentially facilitate the understanding of neural inflammatory responses associated with neurodegeneration.

## Introduction

Macrophages are found in various tissues throughout the body and display a diverse array of morphological and physiological functions. Macrophages are derived from the myeloid lineage and are involved in not only the innate immune system, but also play a crucial role in cell

this manuscript is: Series record GSE171276, located at https://www.ncbi.nlm.nih.gov/geo/info/linking.html. It provides access to all the data associated with this publication.

**Funding:** All authors listed in the manuscript are employees of Fujifilm Cellular Dynamics, Inc. Fujifilm Cellular Dynamics, Inc did not play a role in the study design, data collection and analysis, decision to publish, or preparation of the manuscript and only provided financial support in the form of authors' salaries and/or research materials. Fujifilm Cellular Dynamics provided support in the form of salaries for authors [CAM, SJD, SAB, KB, AS and DR], but did not have any additional role in the study design, data collection and analysis, decision to publish, or preparation of the manuscript.

**Competing interests:** All authors listed in the manuscript are employees of Fujifilm Cellular Dynamics, Inc. The affiliation with Fujifilm Cellular Dynamics Inc, does not alter our adherence to all PLOS ONE policies on sharing data and materials.

**Abbreviations:** iPSC, Induced Pluripotent Stem Cell; SNCA, Alpha-Synuclein; PGRN, Progranulin; MECP2, Methyl CpG binding protein 2; PNSMac, Peripheral Nervous System macrophages; HPC, Hematopoietic Progenitor cell; PD, Parkinson's Disease; GRN, Progranulin Gene; NCL, Neuronal Ceroid Lipofuscinosis; FTD, Frontotemporal Dementia; TDP-43, Transactive response DNA-binding protein of 43 kD; AHN, Apparently Healthy Normal; E8, Essential 8 Medium; EDTA, Ethylenediaminetetraacetic acid; PFA, Paraformaldehyde; SFD, Serum Free Defined; CMP, Common Myeloid Progenitor; BFU-E, Burst Forming Unit-Erythroid; CFU-E, Colony Forming Unit-Erythroid; CFU-GM, Mixed Colony Forming Unit-Granulocyte and Monocyte; CFU-G, Colony Forming Unit-Granulocyte; CFU-M, Colony Forming Unit-Macrophage; LPS, lipopolysaccharide; KO, Knock out; HM, Hemizygous; HO, Homozygous; ROS, Reactive oxygen species; CRF, Control Rate Freezer.

mediated adaptive immunity, development, tissue homeostasis, aid in wound healing, tissue repair, and phagocytosis [1, 2]. Macrophages respond to intrinsic and extrinsic stimuli and secrete growth factors, chemokines, enzymes, along with producing pro- and anti-inflammatory reactions [3]. Tissue resident macrophages are found throughout the body, such as microglia, alveolar macrophages, Langerhans, and Kupffer cells. Besides monocyte derived macrophages circulating in blood, recent papers support the origin and characterization of Peripheral Nervous System macrophages (PNSMacs). PNSMacs are a naturally occurring macrophage population derived from primitive Yolk Sac or definitive Hematopoietic Progenitor cells (HPC). They are located in the endoneurium as well as the epineurium in close proximity to sympathetic neurons and dermal sensory nerves. They reside in many tissues/organs and play a key role in the onset of neurodegenerative diseases [4].

These tissue resident macrophages are localized and respond with corresponding specificity for their tissue niches [5]. Monocyte derived macrophages from blood prove difficult to isolate and propagate. In addition, it is often difficult to procure large volume of blood from specialized clinical centers with necessary consent on a regular basis from different donors, especially from patients with rare diseases. On the same lines it is challenging to isolate PNSMacs from primary donor sources in large numbers. Human Induced Pluripotent stem cell (iPSC) derived macrophages overcome ethical concerns and have a potential to generate limitless supply of reliable and renewable cell types for *in vitro* testing, from normal and disease specific donors. The approach in the current paper was to generate a PNS like macrophage harboring many genetic perturbations observed in various neurodevelopmental (Rett's) or neurodegenerative diseases like Alzheimer's disease (AD) and Parkinson's disease (PD).

The present paper outlines the generation of iPSC and engineered isogenic iPSC lines that mimic mutations associated with Rett- syndrome, Neuronal Ceroid Lipofuscinosis (NCL) and PD by generating isogenic iPSC lines expressing truncated or nonfunctional proteins Alpha-synuclein (SNCA), Progranulin (PGRN), and methyl-CpG-binding protein 2 (MeCP2). These iPSCs are first differentiated to HPCs and then to macrophages. Live and end stage cryopreserved macrophages respond to functional stimuli, retain purity, and phagocytic function. The panel of iPSC derived macrophages offers a unique toolset for studying effects of genetic perturbation on macrophage development in neurodegenerative diseases.

In a healthy brain, macrophages can be easily distinguished by microscopy from microglia by their size and amoeboid morphology. During neuroinflammation, such as in AD, activated microglia and peripheral macrophages alter their respective morphology and marker expression patterns, confounding their distinction. Many neurogenerative diseases, such as AD, exhibit inflammation which can result in peripheral blood macrophages being recruited resulting in alteration of their signature transcripts and morphology similarly to that of microglia [6].

Peripheral macrophages can also act as mediators to enhance the expression of Parkinson's Disease (PD) related genes, such as LRRK2, upregulated by pathogen induced endomembrane damage uncovering a link between membrane damage and onset of PD. Peripheral blood derived macrophages in AD patients with ApoE4/4 genotype reveal impairments in phagocytosing Amyloid beta and are susceptible to apoptosis [7]. Presence of anti-inflammatory M2 macrophages during neural inflammation can contribute to calm chronic inflammation and delay the progression of neurodegenerative diseases. Supplementation of PD patients with low dose of Niacin has been shown to alter the polarization of Macrophages from inflammatory M1 to anti-inflammatory M2 subtype contributing to the suppression of inflammation and halt the progression of PD [8, 9]. During neural inflammation resident and infiltrating macrophages contribute to the secretion of pro inflammatory factors and play a role in increasing neural damage and augment neurodegeneration.

Alpha-synuclein, a 140 amino-acid protein, accounts for about 1% of the total cytosolic proteins in the brain. Its expression is highest in the dopaminergic neurons and is intracellularly localized in presynaptic terminals. Alpha-synuclein proteins can self-assemble, passing from unfolded monomers to oligomeric species, and then to heavy aggregates (called amyloid fibrils). The accumulation of these insoluble fibrils progressively promotes the formation of intracellular inclusions called Lewy bodies within neurons and glial cells [10]. Most of PD cases are sporadic point mutations of SNCA, the gene encoding the Alpha-synuclein protein, which cause inherited forms of PD. The substitution of alanine to threonine at position 53 of the Alpha-synuclein protein (A53T) leads to a severe autosomal dominant trait of Parkinsonism, characterized by an early onset with a short disease duration to death [11, 12].

There has been a strong link between the onset of PD with mitochondrial dysfunction and chronic inflammation. Many immune cell lineages including, T cells, mast cells, and macrophages play a key role in inflammation cascades affecting neural lineages and triggering neural degeneration [13]. Recently Haenseler *et al*. [14] generated iPSC derived macrophages from PD patients to confirm the impairment of normal homeostatic phagocytic function by alpha synuclein and attributing to this mechanism of SNCA in PD patients.

PGRN is a multifunctional growth factor widely expressed in various tissues with highest levels in epithelial and myeloid cells. PGRN is involved in cell proliferation, wound healing and modulation of inflammation [15]. Homozygous mutations in the progranulin gene (GRN) are associated with NCL, a rare lysosomal-storage disorder leading to accumulation of auto-fluorescent lipopigment in lysosomes detectable in various tissues, including skin, retina, and brain. Recently, homozygous GRN mutations were reported to be associated with behavioral variant frontal dementia and Parkinsonism. Heterozygous GRN mutations cause frontotemporal dementia (FTD) with transactive response DNA-binding protein of 43 kD (TDP-43)–positive inclusions [16]. There are no disease-modifying therapies for either FTD or NCL, in part because of a poor understanding of how mutations in GRN contribute to disease pathogenesis and neurodegeneration. This mutation also impairs immune cell types like macrophages. GRN−/− macrophages fail to clear intracellular bacteria due to impairments in autophagy mechanisms. The impairment of autophagy caused by GRN mutations in neurons and immune cells can translate to increased risk of neurodegenerative disease with time [17]. Hence generation of iPSCs derived macrophages harboring GRN mutation can be very valuable to study iPSCs-based disease modeling of modelling for NCL.

MeCP2 is a transcriptional repressor that binds to methylated CpG dinucleotides throughout the genome. Mutations in MeCP2 leading to a partial loss of function are best studied in the onset of Rett syndrome, a neurodevelopmental disorder that affects girls almost exclusively, and in some males [18]. Although Ret syndrome was originally diagnosed exclusively in females, with an approximate prevalence of 1 in 10,000 female births [19], and mutations in the MECP2 gene were previously thought to be fatal for males. A recent review by Volkumar et al. [20] summarized various case reports on the prevalence of MeCP2 mutations in males with Rett syndrome. They observed that nearly 65% of the patients exhibit other genetic co morbidities and hence miss out on the diagnosis for Rett syndrome. The authors emphasize the need to detect, evaluate and treat male patients harboring variant forms of MeCP2 and gather the necessary clinical information on these patient groups. Rett syndrome is characterized by normal early growth and development followed by a slowing of development, loss of purposeful use of the hands, distinctive hand movements, slowed brain and head growth, problems with walking, seizures, and intellectual disability. MeCP2 mutations affect multiple neural cell types like astrocytes, neurons and microglia. In addition to the impairment of function on many neural lineages, peripheral macrophage populations are also sensitive to alterations in MeCP2 expression on a cellular and molecular level highlighting the extensive

crosstalk between the central nervous system and immune system with the onset of Rett syndrome [21]. Although Rett's syndrome is primarily studied in affected females, there are reported cases of Rett's syndrome in males that are severe and lethal [20, 22]. Hence generation of iPSC-derived macrophages and other neural lineages mimicking the loss of function of MeCP2 can be very valuable to study iPSC-based disease modeling of Rett syndrome.

iPSC derived cell lineages are a useful source of patient-specific cells that otherwise are difficult or impossible to derive for disease modeling. In the present paper we have followed through on this approach to interrogate disease associated genotypes in a physiological context by first generating isogenic iPSCs containing a targeted gene mutation mimicking PD (SNCA A53T Homozygous (HO)), NCL (GRN R493X Homozygous (HO)) and Rett Syndrome (MECP2 Hemizygous Truncation (HM) knockout (KO)). Then further differentiating these iPSCs into functional, cryopreservable macrophages, along with an unengineered isogenic parental iPSC donor line as a reference. This panel of macrophages offers a unique toolset for many genotype-specific preclinical therapeutic discovery programs for neurodegenerative diseases. The present paper offers a unique tool set to study the role of peripheral macrophages engineered to express mutations associated typically seen in neuronal cells and understand the effects of these mutations in non-neural cell lineages and their contributions to enable in vitro disease modeling applications for neuro degenerative diseases.

## Material and methods

### Generation of iPSC lines

Multiple human iPSC lines were utilized in these experiments, including an apparently healthy normal (AHN) line and several engineered iPSC lines mimicking disease associated genotypes. The AHN line was the parent line for all engineering, allowing for isogenic models. The AHN human iPSC line (01279) was generated from male PBMCs. Using an episomal reprogramming method, a CD34+ enriched population was electroporated with an EBNA1/oriP based reprogramming vectors [23, 24] then seeded onto RetroNectin-coated plates (Takara Bio, Inc; Otsu, Shiga Japan) and fed using a cocktail of small molecules before being transitioned to TeSR2 medium once iPSC colony formation was observed. The isolated colonies were then selected, propagated and cryopreserved for further characterization, genome engineering and differentiation experiments. This work was previously described by Mack *et al*. [25].

SNCA A53T HO (SNCA A53T) iPSCs were genetically engineered from AHN iPSC 01279 by nuclease-mediated homologous recombination and a donor oligo SJD 14–133. The resulting iPSCs contained SNP rs104893877 where amino acid 53 was changed from alanine to threonine resulting in the A53T variant in the alpha-synuclein gene (SNCA) as well as two silent mutations resulting in the SNCA A53T Homozygous iPSC line (SNCA A53T), provided a disease model for PD.

GRN R493X HO (GRN R493X) iPSC was generated from AHN iPSC 01279 using nuclease-mediated homologous recombination and a donor oligo SJD 14–180. The resulting GRN R493X iPSCs contained SNP rs63751294 changing amino acid 493 from arginine to a termination codon resulting in the early termination of the progranulin gene (GRN) as well as a three base change following the termination codon. The GRN R493X Homozygous diseased line (GRN R493X) provided a model for NCL.

Finally, AHN iPSC 01279 MeCP2 HM KO (MECP2 HM) KO line cell line was generated using nuclease-mediated homologous recombination and a donor plasmid p1553. Donor plasmid p1553 inserted a series of stop codons prior to the Methyl CpG Binding domain followed by a PGKp-PuromycinR-SV40pA selection cassette flanked by LoxP sites. The MECP2 HM KO line provided a disease model for Rett Syndrome.

## Characterization of iPSC lines

iPSC lines 01279, SNCA A53T, GRN R493X, and MECP2 HM were maintained on Matrigel (BD) coated plates in Essential 8 Medium (E8) (Thermo Fisher). Cultures were dissociated using Ethylenediaminetetraacetic acid (EDTA) (Invitrogen) and re-plated onto Matrigel coated plates for a minimum of 5 passages in hypoxia, 5% $CO_2$ and $O_2$ (maintained by addition of nitrogen) prior to differentiation. G-banding analysis was performed by WiCell (Madison, WI) for confirmation of normal karyotype and each experiment was performed within 5 passages of normal karyotype results. All iPSC lines were stained for the presence of pluripotency markers by flow cytometry analysis (Accuri C6, BD). Undifferentiated iPSCs were harvested using EDTA, washed in FACS Buffer (PBS, Invitrogen + 2% FBS, Hyclone), and the cell count was determined. Around 100,000 cells were stained to detect the cell surface expression of TRA-160 (BioLegend, USA clone Tra-1-60-R at 1:200 dilution), SSEA4 (BD Pharmingen, USA clone MC813-70 at 1:20 dilution), and CD90 (BD Pharmingen, USA clone 5E10 at 1:20 dilution). To detect intracellular pluripotency associated proteins the samples were fixed using 4% paraformaldehyde (PFA) (Sigma) for 15 min, permeabilized using 0.1% saponin and stained for the presence of SOX2 (BD Pharmingen, USA clone 030–678 at 1:20 dilution), Oct3/4 (BD Pharmingen, USA clone 40/Oct-3 at 1:5 dilution), and NANOG (BD Pharmingen, USA clone N31-355 at 1:20 dilution) expression by flow cytometry. Confirmation of engineered sequence across the engineered loci was confirmed by Sanger sequence for engineered iPSC lines.

## Differentiation of iPSCs to Hematopoietic Progenitor Cells (HPCs) and end stage macrophages

iPSC lines 01279, SNCA A53T, GRN R493X, and MECP2 HM were maintained on Matrigel or Vitronectin in the presence of E8 and adapted to hypoxia for at least 5–10 passages. The entire hematopoietic differentiation was performed under hypoxic conditions. iPSCs were harvested from sub-confluent iPSCs and the hematopoietic differentiation was initiated by forming aggregates at a density of 0.3–0.5e6cells/mL in the presence Serum Free Defined (SFD) media supplemented with 5 μM blebbistatin (Sigma). 24 hours post aggregate formation, cultures were placed in SFD media supplemented with 50ng/ml zbFGF (Promega), rhBMP4 (R&D Systems), and rhVEGF (R&D Systems) for the first five days. On day 6 of differentiation aggregates were placed in SFD media supplemented with 50ng/ml rhFlt-3 ligand (PeproTech), rhTPO (PeproTech), rhSCF (PeproTech), rhIL-3 (PeproTech), and rhIL-6 (PeproTech) for next 8 days. At the end of the HPC differentiation process, the aggregates were dissociated with TrypLE (Invitrogen) and the emerging cultures were stained for the presence of CD31 (BD Pharmingen, USA clone WM59 at 1:20 dilution), CD34 (BD Pharmingen, USA clone 581 at 1:20 dilution), CD43 (BD Pharmingen, USA clone 1G10 at 1:20 dilution), CD45 (BD Pharmingen, USA clone H130 at 1:20 dilution), CD41 (BD Pharmingen, USA clone HIP8 at 1:20 dilution), and CD235a (BD Pharmingen, USA clone GA-R2 (HIR2) at 1:200 dilution) expression by flow cytometry (BD Accuri C6).

The presence of HPCs were characterized by CD34$^{high}$/CD43$^{high}$/CD45$^{low}$, CD41$^{low}$/CD235a$^{low}$, and CD34$^{high}$/CD31$^{high}$ populations. Purification of HPCs was achieved using Indirect CD34 MicroBead Kit (Miltenyi Systems). The CD34 positive cells were cryopreserved in CryoStor10 (Biolife Solutions) using a Control Rate Freezer (CRF) and transferred to liquid nitrogen tanks for storage. Multipotency of HPCs derived from parental and isogenically engineered lines was confirmed by plating 5000 HPCs on serum free MethoCult (STEMCELL Technologies) media and quantifying colony-forming units per the manufacturer's instructions. The colonies were manually scored after 12 days on incubation. Individualized HPCs

were also plated at 100,000 cells per chamber slides in the presence of serum free MegaCult®-C collagen-based medium (STEMCELL Technologies) to detect megakaryocyte progenitors. After 10 days Megacult cultures were dehydrated, fixed, and stained to detect the presence of Mk-specific antigen GPllb/llla (CD41) on megakaryocytes. All slides were processed according to the manufacturer's instructions.

Cryopreserved or live CD34 positive HPCs were differentiated in to macrophages according to the method described by Choi *et al.* [26, 27] with slight modifications in the protocol. HPCs were first expanded to generate Common Myeloid Progenitors (CMPs) that were stained with CD34, CD43, and CD45 to demonstrate $CD34^{low}/CD43^{high}/CD45^{high}$ population characterizing the emergence of CMPs. CMPs were further differentiated into macrophages using a serum free maturation medium in the presence of M-CSF and IL-1Beta (PeproTech). Cultures were maintained in a normoxic environment, fed every four days and sampled for emergence of CD68 positive cells. Cells were sampled and fixed using 4% PFA for 15 minutes and then permeabilized using saponin. The percentage of CD68 was quantified by flow cytometry. Once end stage cultures reached a high purity of ≥80% CD68 expression, cells were harvested and cryopreserved in CryoStor10 (BioLife Solutions). The vials were cryopreserved using a CRF and transferred to liquid nitrogen tanks for storage. The duration of differentiation from iPSC to end stage macrophages varied from 39–49 days.

## Characterization of macrophages

Characterization of end stage macrophages derived from AHN 01279, and engineered SNCA A53T, GRN R493X, and MECP2 HM, was performed post-cryopreservation. Macrophages were characterized by cell surface expression of macrophage specific markers and presence of distinct morphology. Live and cryopreserved macrophages post thaw were assessed for cytokine release and phagocytosis function. Cryopreserved macrophages were thawed and placed in end stage macrophage maturation media post thaw for 1–3 days to enable staining of cell surface markers, stimulations, and phagocytosis functional assays. Live macrophage cultures were harvested and tested alongside cryopreserved macrophages for all functional assays.

For morphological assessment, macrophages were transferred to Shandon Double Cytoslides (Thermo Scientific) and stained using Wright Stain (Fisher Scientific).

Macrophages were stained to detect the cell surface expression of CD68 (BD Pharmingen, USA clone Y1/82A at 1:20 dilution), CD64 (BioLegend, USA clone 10.1 at 1:20 dilution), CD80 (Biolegend, USA clone 2D10 at 1:20 dilution), CD33 (BioLegend, USA clone WM53 at 1:20 dilution), CD11b (BioLegend, USA clone 3.9 at 1:20 dilution), CD11c (BioLegend, USA clone H130 at 1:20 dilution), CD1a (BioLegend, USA clone HI149 at 1:20 dilution), CD169 (BioLegend, USA clone 7–239 at 1:20 dilution), CD206 (BioLegend, USA clone 15–2 at 1:20 dilution), CD340 (BioLegend, USA clone 24D2 at 1:20 dilution), CD45 (BioLegend, USA clone HI30 at 1:20 dilution), CD163 (R&D Systems, USA clone 215927 at 1:10 dilution), HLA-DR (BioLegend, USA clone L243 at 1:20 dilution), CD86 (BioLegend, USA clone IT2.2 at 1:20 dilution), CD47 (BioLegend, USA clone CC2C6 at 1:20 dilution), TREM2 (R&D Systems, USA clone 237920 at 1:10 dilution), CX3CR (BioLegend, USA clone 2A9-1 at 1:20 dilution), CD38 (BD Pharmingen, USA clone HIT2 at 1:20 dilution), and HLA-ABC (Invitrogen, USA clone 237920 at 1:20 dilution) by flow cytometry using the relevant matched controls.

Live and cryopreserved macrophages were plated at 30,000 cells per well into 96 Ultra Low Attachment (ULA) plate (Corning) in cytokine free base media. Cryopreserved macrophages were placed in media and allowed to rest for three days. The macrophages were then stimulated using LPS (1μg/mL), LPS (1μg/mL) + IFN-Gamma (50ng/mL), IL-4 (50ng/mL) + IL-13 (50ng/mL), IL-10 (50ng/mL) + TGF-Beta (25ng/mL), and TGF-Beta (25ng/mL). Untreated

wells were kept as controls. The supernatants were collected 24 hours after stimulation and frozen at -20˚C for later analysis. The analytes were quantified using a multi-plex Luminex kit and the results were acquired on a FLEXMAP 3D system (Luminex). Data analysis was carried out using the xPONENT Multiplex Assay Analysis software (Luminex).

The phagocytotic capability of the live and cryopreserved macrophages were measured to access functionality. Macrophages were plated at 15,000 cells per well in tissue culture (TC) 96 well plate (Corning) in SFD media supplemented with ExCyte (Millipore), 20ng/ml M-SCF (PeproTech) and 10 ng/mL IL-1Beta (PeproTech). After 24 hours the wells were treated with pHrodo labeled Red *S. aureus* bioparticles (Thermo Fisher) at 0.5 µg/well, placed into IncuCyte S3 and measured for red fluorescent intensity for 6 days. As the *S. aureus* bioparticles are internalized the pHrodo dye reacts with the decrease of pH resulting in increasing level of fluorescence. Phase and red fluorescence images were captured in the IncuCyte S3 every 2 hours for 6 days at 10x magnification. Phagocytosis was quantified using IncuCyte® s3 software (v2019B), which applied a pre-defined analysis definition to remove background fluorescence, allowing accurate quantification of red fluorescence intensity.

## Sequence analysis

Genomic DNA (gDNA) was extracted from the iPSC lines and the differentiated cells post thaw. The regions of interest were amplified and submitted to Functional Bioscience (Madison, WI) for sequencing. The resulting sequence data were aligned to the expected sequence maps using SeqMan Pro (DNAStar, Madison, WI).

## Human macrophage preparation, RNA-Seq library preparation, sequencing, and data analysis

Briefly, parental and isogenically engineered iPSCs were differentiated to end stage macrophages expressing >80% CD68 positive cells. The cells were cryopreserved and stored in the presence of liquid nitrogen. RNA seq analysis was performed on macrophages three days post thaw maintained in the presence of macrophage medium. RNA was extracted with RNeasy Mini Kit (Qiagen) according to the manufacturer's protocol. After extraction, the sample was incubated with Turbo DNase at 37˚C for 30 minutes and subsequently re-purified using RNeasy clean-up protocol.

RNA sequencing was performed on Illumina NovaSeq 6000 platform (Illumina Inc., San Diego, CA) at Novogene, China targeting at least 20 million paired-end reads with a read length of 150 bp. Reads were mapped to the Ensembl (GRCh38.p10) Homo sapiens genome using splice-aware alignment program, HISAT2, v.2.1.0. Differential gene expression analysis between the wild-type and mutant macrophage samples was carried out using Cuffdiff, v.2.2.1 program. Custom perl scripts were used to convert the Cuffdiff output files to compatible text files for visualization in TIBCO Spotfire software v.11.0.0. Visualization of the mapped reads was accomplished using the Integrative Genomic Viewer, v. 2.4.15 (http://www.broadinstitute.org/igv). The GO biological process functional enrichment analysis of the statistically significant DEGs obtained from the RNASeq analysis between the mutant MO and the WT was accomplished in R using ClusterProfiler R package (http://www.bioconductor.org/packages/release/bioc/html/clusterProfiler.html). The hypergeometric distribution method was used to calculate the P-value. P<0.05 was used as the threshold for statistical significance. The results of ClusterProfiler enrich analysis were visualized via the enrichplot (https://github.com/GuangchuangYu/enrichplot) and cnetplot package [28, 29]. Transcription tracks were assembled utilizing Trinity v2.8.4 assembler using default settings was used to assemble the transcriptomes *de novo for* both the 01279 MO and MeCP2 HZ MO mutant lines. All the 01279

and mutant transcripts which matched the HG38 MeCP2 sequence were identified using BLAST. A multiple sequence alignment of the assembled transcripts showing the disrupted mutant exon region was performed using clustalW.

## Results

### Generation of integration-free iPSC lines from AHN donor and subsequent engineering strategies to generate isogenic iPSC lines expressing SNCA A53T, GRN R493X, and MeCP2 HM

AHN iPSC line 01279 generated using episomal reprogramming methods previously described by Mack *et al.* [25] was used for generating all engineered isogenic lines. AHN 01279 iPSCs were transfected using nuclease-mediated homologous recombination to generate three lines, SNCA A53T (Fig 1A), and GRN R493X (Fig 1B) and MECP2 HM (Fig 1C). Nuclease design considerations included pseudogenes, repetitive sequences and potential "off-by-one" cut sites in the genome. The SNCA A53T and GRN R493X nuclease recognition sites were analyzed using TAGScan (https://ccg.epfl.ch/tagger/tagscan.html) [30]. Two off-by-one sites were identified for the SNCA A53T nuclease and one site for the GRN R493X nuclease. Sanger sequence across these sites was performed on the engineered iPSC lines to confirm the sequence at these

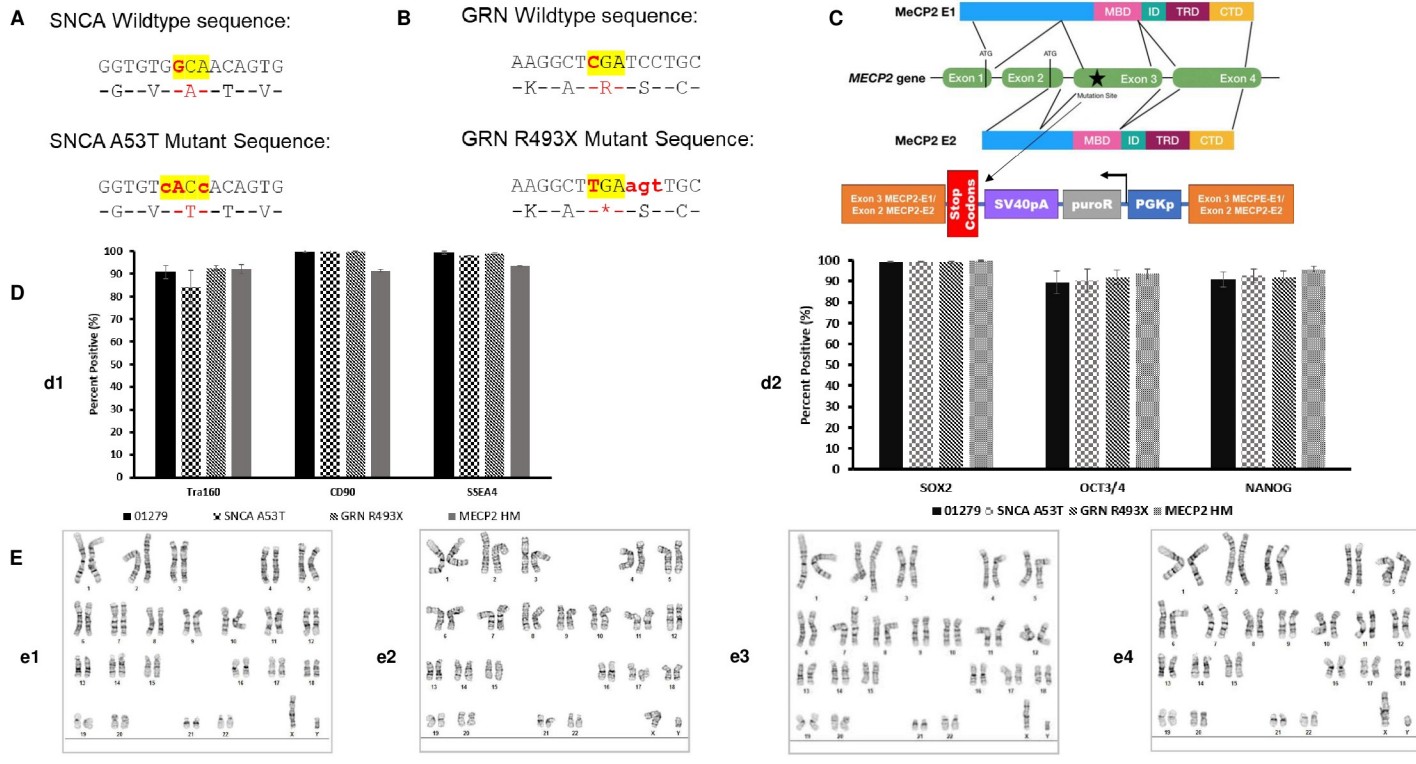

**Fig 1. Generation and characterization of isogenic iPSCs for disease modelling.** (A) SNCA A53T engineering is schematically described, showing the amino acid 53 change from alanine to threonine highlighted in yellow. (B) GRN R493X engineering is schematically described, showing the amino acid change of arginine 493 to a termination codon highlighted in yellow. (C) MECP2 HM KO engineering is schematically described, showing insertion of stop codons prior to Methyl CpG Binding domain followed by a PGKp-PuromycinR-SV40pA selection cassette. (D) iPSC characterization preformed using flow cytometry for the presence of traditional pluripotency markers. (d1) Detections of surface markers Tra160, CD90, SSEA4, and (d2) intracellular markers SOX2, Oct3/4, and Nanog. Each graph quantifies percentage of pluripotency markers (n = 4 ± SE) for 01279, SNCA A53T, GRN R493X, and MECP2 HM. Parental and engineered cell lines all displayed surface and intracellular markers confirming a pluripotent state. (E) Cytogenetic analysis on G-banded metaphase cells from each iPS cell line detailing a normal karyotype, (e1) 01279, (e2) SNCA A53T, (e3) GRN R493X, and (e4) MECP2 HM. G-banding karyotyping preformed and analyzed by WiCell (Madison, WI).

loci matched the starting, unengineered cell line (S1 Fig and S1 Table). MECP2 TAL design was analyzed by ZFN-site (https://ccg.epfl.ch/tagger/targetsearch.html [30]) there were no predicted off-target sites with 2 or fewer mismatches per half site. The selected iPSC clones and parental lines were maintained using Matrigel coated plates and E8 media under hypoxic conditions for at least 5 passages and then cryopreserved. The parental iPSC line and genetically engineered iPSC lines were characterized further by quantifying the high purity of pluripotency markers TRA-160, SSEA-4, CD90, SOX-2, OCT3/4 and Nanog expression by flow cytometry (Fig 1D1 and 1D2 and S2 Table). The iPSCs banks displayed a normal karyotype (Fig 1E1–1E4) by G-banding karyotype analysis. Each of these engineered iPSCs offered an opportunity to study the effect on of selected mutations on hematopoiesis and differentiation to macrophages.

## Parental and isogenic engineered iPSCs can differentiate to Hematopoietic Progenitor Cells (HPCs) at varying efficiencies

Hematopoietic differentiation was initiated by harvesting the iPSC cultures and generating aggregates (3D) under defined serum free conditions. A schematic of the differentiation process is displayed in Fig 2A. The protocol was able to successfully generate HPC from parental and isogenically engineered lines. The key phenotypic markers observed during hematopoietic differentiation, which are absent on iPSC (S2B Fig), are CD43+/CD34+ cells representing HPCs, CD41+/CD235+ cells representing erythromegakaryocytic lineage, CD45+/CD43+ representing myeloid lineages. Parental and isogenically engineered iPSC lines were placed in hematopoietic differentiation for ~13 days and the expression profile of all hematopoietic markers was quantified by flow cytometry. The expression profiles, of hematopoietic lineage specific markers are depicted in Fig 2B. Although all iPSC successfully differentiated to HPCs, differences were observed in the expression levels of many markers. MECP2 HM iPSC revealed the highest expression of HPC associated markers on day 13 of differentiation as compared to SNCA A53T and GRN R493X engineered iPSCs. MECP2 HM IPSC derived HPC revealed the highest expression of erythroid lineage specific markers, while SNCA A53T iPSC derived HPCs revealed the lowest levels of hematopoietic, erythroid and myeloid specific markers (Fig 2C). The ability of HPCs to differentiate into megakaryocytes and platelets was determined by placing HPCs in a Collagen IV based CFU assay for megakaryocyte progenitors for 10 days. The colonies were stained for the presence of CD41 and confirmed the megakaryocytic potential of HPCs (Fig 2D). The multipotency of HPCs was assessed using colony-forming unit (CFU) assays to assess the proliferation and differentiation of HPCs towards different downstream lineages. CFU-E (colony-forming unit-erythroid) and the BFU-E (burst-forming unit-erythroid) scored for erythroid cells; CFU-G (colony-forming unit-granulocyte) CFU-M (colony-forming unit-macrophage), CFU-GM (colony-forming unit-granulocyte/macrophage) scored for the myeloid cell type while the CFU-GEMM (colony-forming unit-granulocyte/erythroid/macrophage/megakaryocyte) scored for the presence of true multipotent colonies.

CFUs plates from 01279 (n = 32), SNCA A53T (n = 6), GRN R493X (n = 6), and MECP2 HM (n = 6) derived HPCs were manually scored and quantified (Fig 2E and S3 Table). The functional assessment of HPCs derived from parental and isogenically engineered iPSCs revealed the lowest number of immature BFU-E (burst forming unit-erythroid) units in MECP2 HM derived HPCs indicating that MeCP2 may have role in erythroid maturation. MECP2 HM iPSC derived HPCs also showed an increase in the granulocyte/monocyte population compared to the parental unengineered iPSCs. SNCA A53T iPSC derived HPCs revealed a lower CFU potential for all lineages compared to the parental iPSC derived HPCs

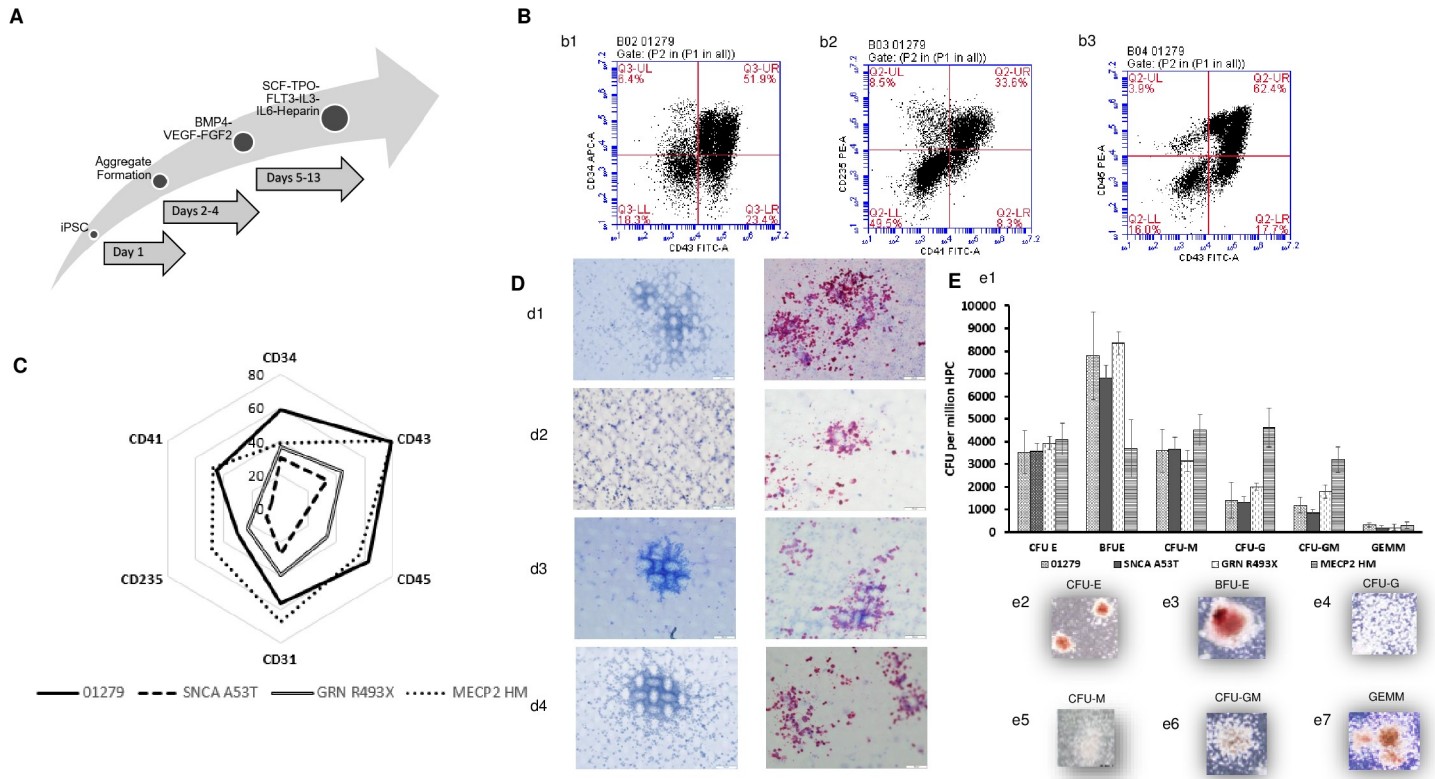

**Fig 2. Generation and characterization of HPCs from isogenic iPSCs for disease modelling.** (A) A schematic representation of the differentiation process from iPSC to HPC with the timeline and cytokines utilized throughout the process. (B) Representative flow cytometry dot plots for the HPCs' (b1) lymphoid, (b2) erythroid, and (b3) myeloid potential from 01279 iPSCs. (C) Quantification of HPC purity was confirmed by detection of, CD34, CD43, CD41, CD235a, and CD45 expression on HPCs derived from 01279, SNCA A53T, GRN R493X, and MECP2 HM is depicted. (D) Megakaryocytes/proplatelets stained using MegaCult-C kit. HPCs were plated in collagen-based assay (MegaCult) and to generate megakaryocytes capable of shedding pro-platelets. The emerging megakaryocytes were stained for the presence of CD41 or a matched isotype control and the staining was captured by alkaline phosphatase staining. Presence of megakaryocytes/proplatelets was confirmed by detection of glycoprotein IIb/IIIa accompanied by visualization of pink color. Representative isotype and megakaryocytes/proplatelets images captured using 20X magnification, (d1) 01279, (d2) SNCA A53T, (d3) GRN R493X, and (d4) MECP2 HM. (E) (e1) Multipotency of HPC quantified using the serum free methocult Colony Forming Unit (CFU) assay. The presence of erythroid (CFU-E/BFU-E), myeloid (CFU-M), granulocyte (CFU-G), granulocyte-macrophage (CFU-GM), and multipotent granulocyte-erythroid-macrophage-megakaryocyte (CFU-GEMM) were scored manually and the graphs represent average ± SE for each line. The graph depicts CFU from 01279 (n = 32± SE), SNCA A53T (n = 6± SE), GRN R493X (n = 6± SE), and MECP2 HM (n = 6± SE). A representation of emerging CFU colonies, CFU-E (e2), BFU-E (e3), CFU-G (e4), CFU-M (e5), CFU-GM (e6), and GEMM (e7).

while the CFU potential of GRN R493X iPSC derived HPCs remained comparable to the parental unengineered iPSCs (Fig 2E). This the first report describing the differences in hematopoietic potential in iPSC derived lines, SNCA A53T, GRN R493X and MECP2 HM HPCs.

## Pure end stage macrophages derived from iPSCs from parental and isogenic engineered iPSCs

HPCs derived from parental and isogenically engineered-iPSCs were further differentiated to CMP (Fig 3A) and phenotypically characterized by the by loss of CD34 (Fig 3B) with concomitant increase in a cell population co-expressing CD43 and CD45. The emerging CMPs retain the potential to generate multiple myeloid lineages (mast cells, dendritic cells, granulocytes) besides generating end stage macrophages. The CMPs were expanded for 8 days under normoxic conditions in a 3D culture format. The enrichment and maturation to end stage macrophages was driven by M-CSF and IL-1 beta [26]. An in-process purity assessment of end stage

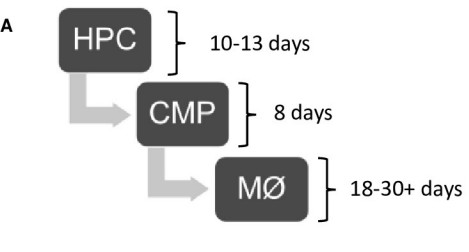

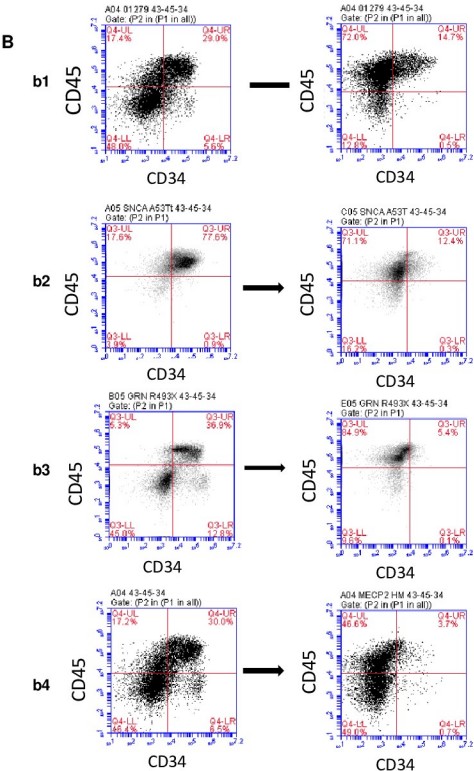

**Fig 3. Generation and characterization of isogenic iPSCs for disease modelling.** (A) Schematic representation of the differentiation process from HPC to common myeloid progenitor (CMP) to macrophages under defined conditions. (B) Loss of CD34 expression and augmentation of HPC and CMP stage of differentiation for (b1) 01279, (b2) SNCA A53T, (b3) GRN R493X, and (b4) MECP2 HM KO. (C) Efficiency of generating macrophage from iPSCs lines. The efficiency calculated by total viable cell number of iPSC seeded and total number viable macrophages obtained at the end of the process.

cultures was performed by quantification of CD68 expression. As the purity of CD68 positive cells increased to >80% the cultures were harvested and cryopreserved. End stage macrophages were successfully generated from parental and isogenically engineered iPSCs. The efficiency of generation of macrophages varied between iPSC lines. The parental 01279 cells revealed the highest efficiency of generating 1.74 macrophages per iPSC, followed by SNCA A53T and GRN R493X, while MECP2 HM revealed the lowest efficiency of generating 0.21 macrophages per iPSC (Fig 3C).

### Cryopreserved macrophages derived from parental and isogenic engineered iPSCs exhibit features of unpolarized/naïve macrophages

The identity of iPSC-derived macrophages was established by the qualitative analysis of cellular morphology and quantification of relevant phenotypic markers by flow cytometry. Macrophages are distinguished from other types of myelomonocytic cells by the distinct morphology observed by Wright staining and staining for a panel of surface markers. Cryopreserved macrophages exhibited high viability and recovery post thaw (S3 Fig). Cryopreserved macrophages from parental and isogenically engineered iPSCs revealed the classic macrophage morphology by Wright staining (Fig 4A). Expression of CD68 post thaw was evaluated to determine purity of macrophages derived from parental and isogenically engineered lines (Fig 4B). Macrophages were also stained for an extended panel of phenotypic markers three days post thaw (Fig 4C and S4 Table). Pro-inflammatory (M1) macrophages, characterized by the secretion of

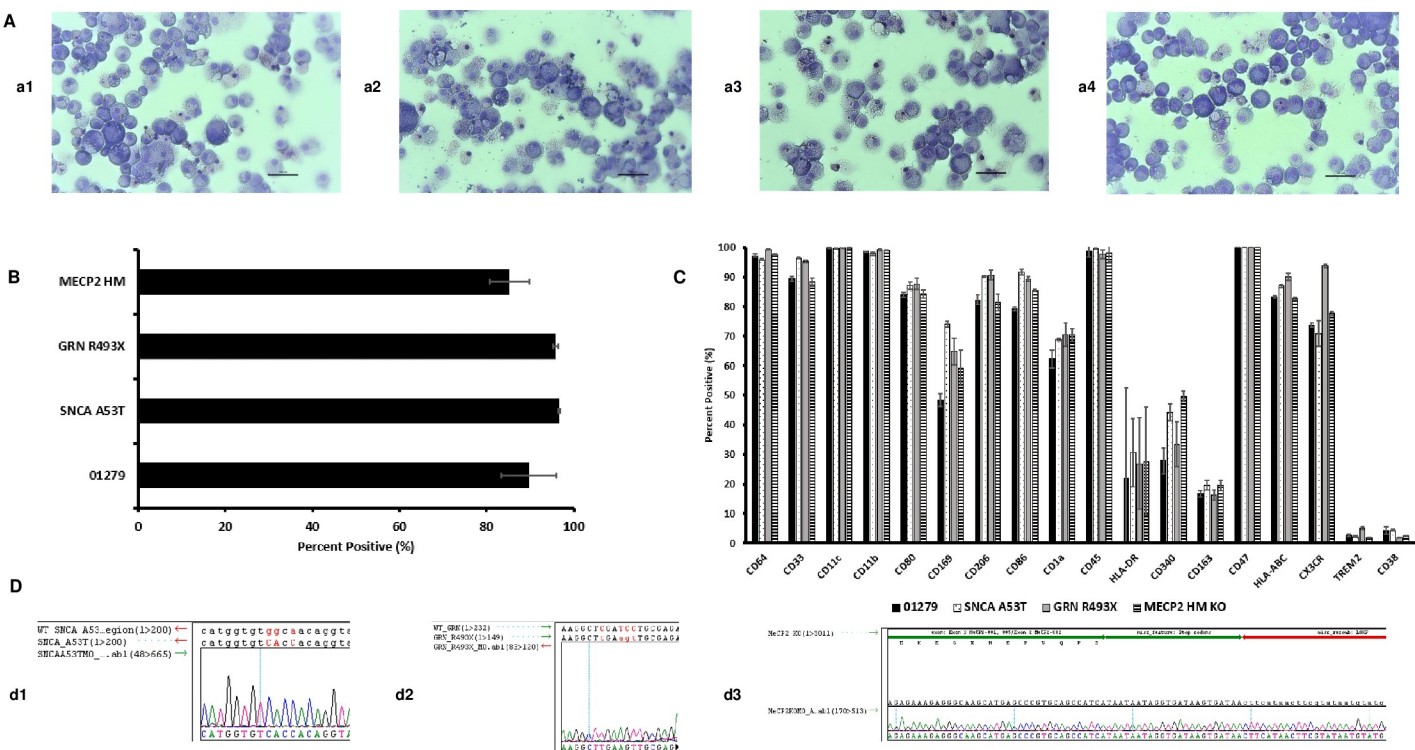

**Fig 4. Functional characterization and sequencing of macrophages from isogenic iPSC for disease modelling.** (A) Morphological analysis post thaw was performed by staining the cultures using wright stain (a1) 01279, (a2) SNCA A53T, (a3) GRN R493X, and (a4) MECP2 HM. All pictures captured at 20X objective. (B) Quantification of CD68 expression post thaw by flow cytometry. The graphs depict an average of 3 samples ± SE. (C) Additional characterization of the macrophages by staining cells for the presence of CD64, CD33, CD11c, CD11b, CD80, CD169, CD206, CD86, CD1a, CD45, CD340, CD163, CD47, HLA-ABC, CX3CR, TREM2, and CD38 expression by flow cytometry. The graphs depict and average of 3 samples ± SE. (D) Sequence confirmation of genetically engineered iPSC Derived macrophage. (d1) SNCA A53T macrophage sequence (d2) GRN R493X macrophage sequence (d3) MeCP2 HM KO macrophage sequence. Alignment was performed to the wildtype sequence (WT) and appropriate expected mutation sequence for d1 and d2. The alignment was performed to the expected inserted sequence alone for MeCP2 HM (d3).

pro-inflammatory cytokines, high expression of CD64, CD80, CD86 and low expression of CD206 are responsible for phagocytosis and ADCC killing. Anti-inflammatory (M2) macrophages, which are involved in tissue repair and provide regulatory signals, express high levels of CD206 and CD11b.

End stage macrophages from parental and engineered lines revealed high expression of myeloid specific markers, CD33 (Siglec-3), CD11c; a member of the leukointegrin family, which binds to complement fragment (iC3b), and pan macrophage sialomucin CD68, a member of the scavenger receptor supergene family. Cryopreserved iPSC-derived macrophages also expressed major histocompatibility complex class II (HLADR), co-stimulatory molecules CD80 and CD86 [31] that support antigen presentation role for macrophages, high affinity immunoglobulin gamma Fc receptor CD64 that plays a central role in macrophage antibody-dependent cellular cytotoxicity (ADCC) and release of pro-inflammatory cytokines expressed on M1 macrophages. Cryopreserved IPSC-derived macrophages also constitutively expressed high levels of CD206 a MRC1, a C-type mannose receptor 1 involved in immune homeostasis by scavenging unwanted mannoglycoproteins and CD11b, a β2 integrin involved in adhesion, which binds to inactivated complement 3b, expressed on M2 macrophages and microglia [32]. iPSC derived macrophages expressed high levels of MHC class I expression and low levels of CD1a expression which could aid in immune recognition and antigen presentation function. IPSC-derived macrophages also expressed CD169; a marker of a subpopulation of

macrophages found in lymphoid organs and implicated in immune tolerance and antigen presentation [33]. Parental and isogenically engineered macrophages did not have detectable expression of human dendritic cell specific marker CD209 (DC-SIGN). The combined expression of both M1 and M2 specific markers indicate an uncommitted / M0 status of iPSC-derived macrophages.

Although end stage macrophages derived from SNCA A53T, GRN R493X and MECP2 HM iPSCs expressed high levels of CD68, there were differences in the levels of expression of additional panel of markers. Comparative levels of expression of M1 (CD11c, CD64, CD80, CD86, and HLA-DR), M2 (CD206 and CD11b) and tissue specific regulatory macrophages (CD169) were observed in parental, SNCA A53T and GRN R493X iPSC derived macrophages. MECP2 HM iPSC derived macrophages revealed significantly lower level of expression of CD33, CD64, CD11b, CD206, CD86 compared to the parental macrophages.

Parental and SNCA A53T, GRN R493X and MECP2 HM macrophages expressed high levels of CD47, 'marker-of-self' protein, is emerging as a novel potent macrophage immune checkpoint for cancer immunotherapy [34]. Parental and SNCA A53T, GRN R493X and MECP2 HM macrophages also expressed high levels of fractalkine receptor CX3CR1, that plays a key role in macrophage maturation, wound healing, and orchestrating immune response [35, 36]. This is the first report presenting confirming the expression of CD47 and CX3CR1 on iPSC derived macrophages implying a role in evading immune response.

Macrophages express CD38 under inflammatory conditions [37]. The absence of CD38 expression on parental, SNCA A53T, GRN R493X and MECP2 HM macrophages confirms the naïve and non-inflammatory status of end stage iPSC derived macrophages. Parental, SNCA A53T, GRN R493X and MECP2 HM macrophages also expressed low levels of Triggering receptor expressed on myeloid cells-2 (TREM-2) a transmembrane immune receptor. The low expression of TREM2 aids to distinguish iPSC derived macrophages from iPSC derived microglia, which express very high levels of TREM2 ([38], S2C Fig).

## Phagocytic function of macrophages is retained pre and post cryopreservation

Phagocytosis is a key feature of macrophage function in host defense and tissue homeostasis. Quantification of macrophage phagocytosis in vitro has been recently made possible utilizing the IncuCyte S3 real time imaging platform and pHrodo labeled pathogen bioparticles, which only fluoresce when localized in the acidic environment of the phagolysosome [39]. Though most of functional analysis has been demonstrated with live cells we were keen to test this function post cryopreservation of end stage macrophages. End stage live and cryopreserved macrophages derived from parental and isogenic engineered iPSCs were able to phagocytose pHrodo labelled *S. aureus*, but there were differences in the kinetics of phagocytosis between lines, fresh and cryopreserved. Live end stage macrophages derived from the parental iPSC revealed a stronger intensity of phagocytosis when benchmarked against cryopreserved end stage macrophages derived from the same parental iPSC (Fig 5). This phenomenon has been reported by other investigators [40] in normal healthy macrophages as a response to cryopreservation process. On the other hand, cryopreserved macrophages from engineered iPSCs revealed a higher intensity of phagocytosis post thaw than live end stage macrophages derived from the same engineered iPSCs (Fig 5).

Side by side comparison of all live end stage macrophages cultures derived from SNCA A53T iPSC, GRN R493X iPSC, and MECP2 HM iPSC revealed a strong impairment of phagocytic function when compared to live end stage parental iPSC derived macrophages (S4 Fig). This observation supports data published by Haenseler *et al.* [14] on the impairment in

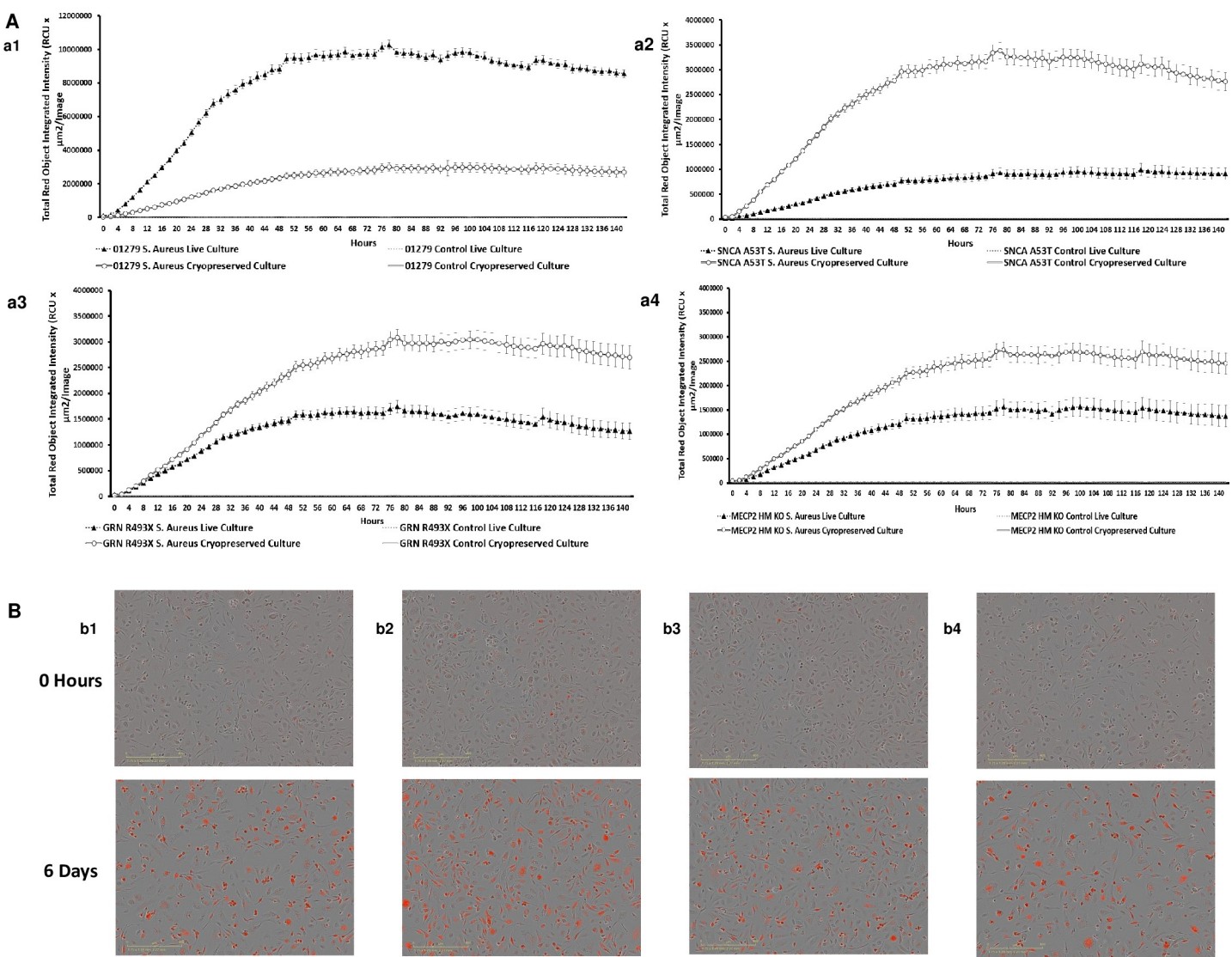

**Fig 5. Phagocytotic/functional ability of iPSC derived macrophages for disease modelling.** (A) The functionality of the macrophages assessed by their ability to phagocytose pHrodo labeled opsonized *S. aureus*. The plated macrophage cultures were monitored to capture an increase in red fluorescent intensity over a 6-day period. Images captured on the IncuCyte S3 every 2 hours and the Total Red Object Integrated Intensity (RCU x $\mu m^2$/image) was measured and analysed using IncuCyte Software (v2019B), correlating to active phagocytosis of the pHrodo *S. aureus* bioparticles by the live (n = 3 ± SE) and cryopreserved (n = 19 ± SE) macrophages derived from 01279 (a1), SNCA A53T (a2), GRN R493X (a3), and MECP2 HM (a4) respectively. (B) Overlay of phase and red fluorescence images demonstrating phagocytosis of pHrodo labeled *S. aureus* bioparticles at 0 hours and 6 days post plating of cryopreserved and thawed macrophage cultures derived from 01279 (b1), SNCA A53T (b2), GRN R493X (b3), and MECP2 HM (b4). Images (10X) generated by IncuCyte s3 System.

phagocytic functions in live iPSC-derived macrophages from PD patients with SNCA Triplication and A53T mutations. Additionally, Wang *et al*. [41] reported an enhanced recruitment of live macrophages with the decreased bacterial clearance, impaired endocytosis capacity of macrophages accompanied with Progranulin (PGRN) deficiency and finally, Cronk *et al*. [42] reported on the differential gene expression profile on live Mecp2-null peritoneal macrophages implying a key role of this gene in sustaining macrophage function [42].

On the other hand, phagocytosis from cryopreserved macrophages generated from SNCA A53T iPSCs revealed a robust phagocytosis when compared to the parental unengineered macrophages. GRN R493X iPSC derived macrophages displayed a similar kinetic trend of

phagocytosis like the parental macrophages. Finally, macrophages derived from MECP2 HM iPSC revealed a striking impairment of phagocytic function when compared to the parental iPSC derived macrophages (S4 Fig). This data highlights the role of MECP2 in regulating phagocytic function in both live and cryopreserved conditions.

A possible explanation to support this observation is the bias towards an M1 sub type in engineered macrophages before cryopreservation due to the high secretion levels of the pro inflammatory cytokines IL-8, CCL11, and CCL17 when compared to the unstimulated parental macrophages in end stage live cultures (Fig 5). The bias towards an M1 polarization could influence metabolic reprogramming in engineered macrophages which in turn could trigger interplay between metabolic enzymes and metabolites of different pathways and contribute to enhanced phagocytosis function in engineered macrophages post cryopreservation [43].

## Live and cryopreserved macrophages elicit functional response to M1 and M2 stimuli

Macrophages acquire functional and polarization signatures during maturation, Mantovani *et al.* [44] grouped the stimuli generated by macrophages in a continuum between two functionally polarized states M1 and M2, based on effects of selected macrophage markers. An M1 or anti-angiogenic or inflammatory response included stimulation with interferon-gamma (IFN-γ) + lipopolysaccharide (LPS) or tumor necrosis factor (TNF) and M2 or anti-inflammatory or pro-healing response was subdivided to accommodate similarities and differences between interleukin-4 (IL-4) (M2a), immune complex + Toll-like receptor (TLR) ligands (M2b), and IL-10 and glucocorticoids (M2c) [45]. In addition, M1-like polarized macrophages exhibit a high level of phagocytic activity, and markers that best characterize them were CD64 and CD80, although the level of expression of these two markers was mainly dependent on the nature of the M1 stimulus (IFN-γ versus LPS versus IFN-γ and LPS). In addition, M1 macrophages secrete proinflammatory cytokines such as IL-1β, IL-6, IL-12, IL-18 and IL-23, TNF-α, and type 2 IFN; and several chemokines such as CXCL1, CXCL3, CXCL5, CXCL8, CXCL9, CXCL10, CXCL11, CXCL13, and CXCL16; CCL2, CCL3, CCL4, CCL5, CCL8, CCL15, CCL11, CCL19, CCL20, and CX3CL [46, 47]. The M2-macrophage plays a role during parasitic, helminthic, and fungal infections and reveal expression for CD64 and CD209, a C-type lectin and induce secretion of IL-13, CCL1, CCL2, CCL13, CCL14, CCL17, CCL18, CCL22, CCL23, CCL24, CCL26, IL-8, monocyte chemo-attractant protein-1 (MCP)-1, IP-10, macrophages inflammatory protein (MIP)-1β, and CCL5 (RANTES).

To analyze the cytokine profile in parental and engineered macrophages, end stage live and cryopreserved macrophages three days post thaw were stimulated with LPS, LPS + interferon, IL-4 + IL-13, IL-10 + TGF-beta, and TGF-beta. The supernatants were examined for the production of key cytokines using the Luminex Multiple assay system (S5–S8 Figs). The fold induction of the analytes in the live engineered macrophages was measured as a fold induction with respect to the analytes released by live parental and engineered macrophages. Similarly, the fold induction of the analytes in the cryopreserved engineered macrophages was measured as a fold induction with respect to the analytes released by cryopreserved parental un engineered macrophages.

Unstimulated parental and engineered iPSC-derived macrophages released similar levels of IL-23, MMP-9, CCL3, CD163, CCL22, and CXCL1 (Fig 6). Unstimulated engineered macrophages had a higher secretion of IL-8, CCL11, and CCL17; and lower levels of IL-1ra, CCL24, and fractalkine compared to the unstimulated parental macrophages. Unstimulated engineered iPSC derived macrophages also spontaneously produced CCL1 in cultures. CCL1 is involved in inflammatory processes through leukocyte recruitment and could play a crucial

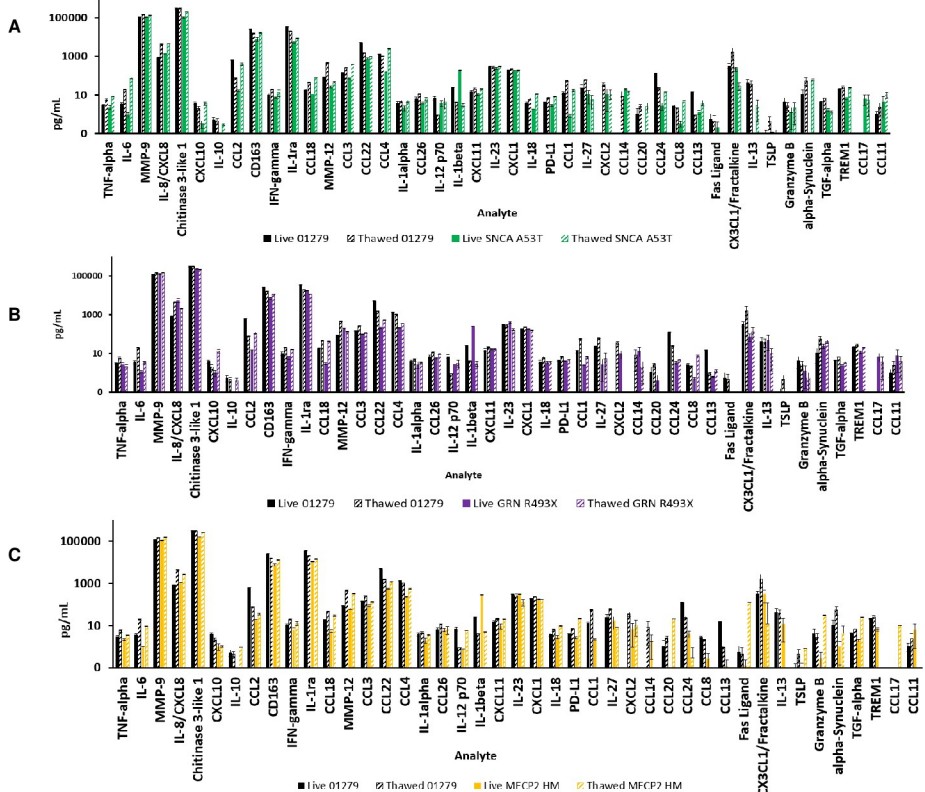

**Fig 6. Polarization studies with iPSC derived macrophages from isogenic iPSC for disease modelling.** Live end stage culture of macrophages as well as cryopreserved macrophages post thaw were allowed to recover for 72 hours in macrophage medium and placed in cytokine free medium overnight prior to addition of stimulants. Macrophages were stimulated with LPS, LPS + Interferon Gamma, IL-4 + IL-13, IL-10 + TGF-Beta, and TGF-Beta for 24 hours. Supernatants were collected, and release of cytokines was quantified by the Luminex multiplex system. The graphs depict the quantification of analytes released by unstimulated parental and engineered iPSC derived macrophages. Each value depicts the average of three samples ± SE. SNCA A53T (6a), GRN R493X (6b), and MECP2 HM (6c).

role in calcium flux, angiogenesis and along with CCL2 and CCL9 are negative regulators of M2 polarization neuroinflammatory disorders [48–50].

The results revealed that end stage live cultures of macrophages secreted lower levels of TNF-alpha, IL-6, PD-L1, IL-8, IL-23, Chitinase 3-like 1,CXCL10, IL-10, CCL2, CXCL2, CCL14, IFN-gamma, IL-1ra, CCL18, CCL24, CCL3, CCL22, CCL4, CCL20, CD163, CCL26, CX3CL1/Fractalkine, IL-13, TSLP, MMP-9, IL-1alpha, MMP-12, IL-12, Granzyme B, SNCA, CXCL11, TREM-1, CCL17, IL-23, CCL11, and IL-18 compared to the cryopreserved macrophages. The one exception was the constitutive high secretion of IL-1 β in the live cultures when compared to the cryopreserved macrophages.

The overall trends of the analytes released are as follows. The live engineered macrophages had comparable levels of TNF-alpha, IL-6, CD163, IGN-gamma, CCL3, CCL4, IL-1alpha, CCL26, CXCL1, IL1-beta, CXCL12, CCL24, and CCL18. Cryopreserved SNCA macrophages released higher levels of TNF-alpha, IL-6, IL-8, CXCL10, CCL2, CCL18, CCL3, CCL4, IL-23, IL-18, and CCL18 (Fig 7). Cryopreserved SNCA A53T macrophages released lower levels of IL-10, MMP-12, CCL1, TSLP, TGF-alpha, TREM1, and absence of Fas ligand. Cryopreserved GRN R493X macrophages had the highest levels of CXCL10 compared to parental and SNCA A53T and MECP2 HM macrophages. Cryopreserved GRN R493X macrophages also secreted higher levels of CCL2 and lower levels of TGF-alpha, Granzyme B, TSLP, IL-13, CCL14,

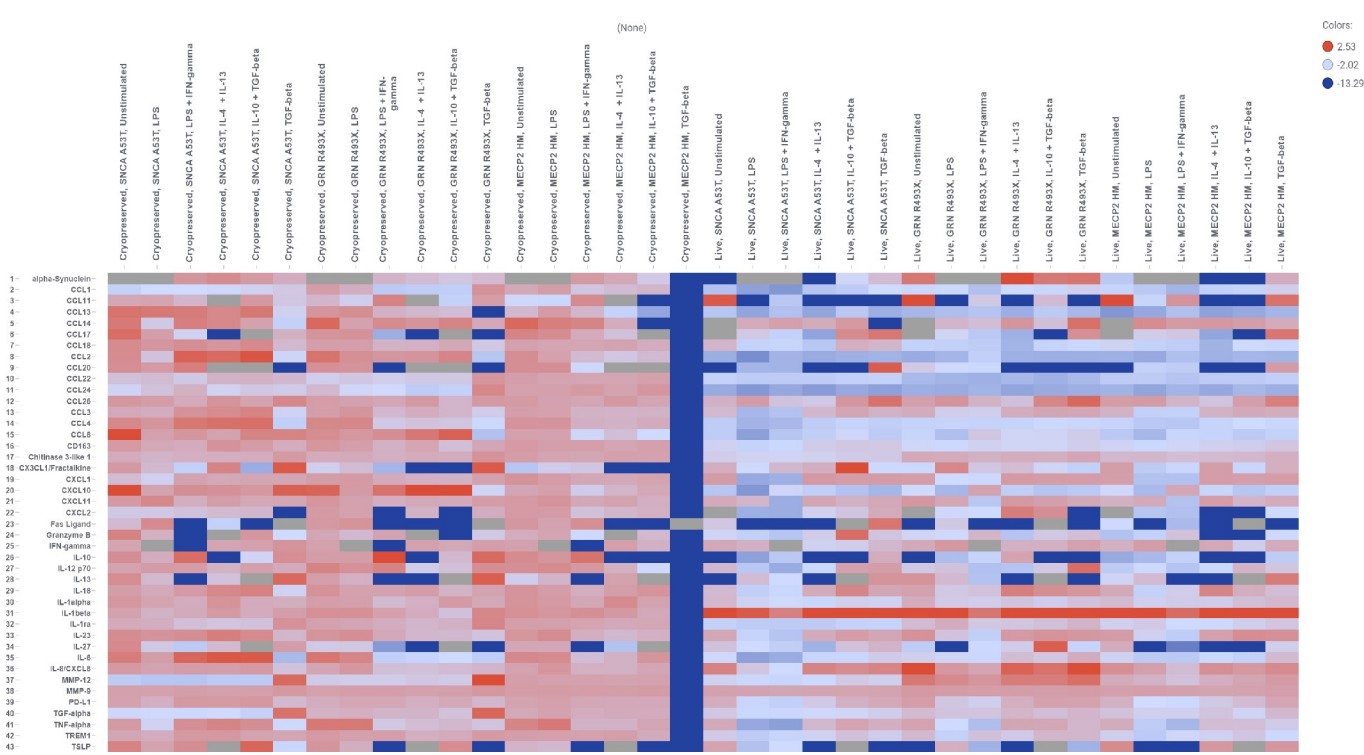

**Fig 7. Polarization studies with iPSC derived macrophages from isogenic iPSC for disease modelling.**

CCL1, PDL1, IL-18, CCL4, and the lowest levels of CCL2. Cryopreserved MECP2 HM macrophages released higher levels of IL-10, IL-18, PDL1, CCL20, TSLP, Granzyme B and, the highest level of Fas ligand. MECP2 HM cryopreserved macrophages had an absence of TREM1, IL-13, CCL18, and CCL14. Cryopreserved MECP2 HM macrophages released lower levels of IL-6, TNF-alpha, IL-8, CXCL10, CCL2, CD163, CCL3, INF-alpha, IL-23, and alpha-synuclein.

Cryopreserved MECP2 HM macrophages when treated with TGF beta revealed a steep downregulation of all analytes when compared to end stage live unstimulated MeCP2HM macrophages [51] implying a key role for the BMP4/TGF-beta signaling pathways associated with perturbations of MeCP2. Unstimulated cryopreserved MeCP2 HM macrophages secreted less IL-13 than cryopreserved unstimulated cryopreserved SNCA A53T and GRN R493X macrophages. Presence of IL-13 and IL-4 have been shown to reduce inflammation by promoting the M2 microglia phenotype and contributing to the death of microglia M1 phenotype, or by potentiating the effects of oxidative stress on neurons during neuro-inflammation [52]. These results revealed that the cryopreserved engineered macrophages were responsive to both M1 and M2 stimuli and secreted relevant analytes (Fig 7).

Live and cryopreserved macrophages were plated in cytokine free media and cytokine release was quantified per stimulant. Cryopreserved macrophages were thawed and allowed to recover for 72 hours prior to stimulation. Macrophages were stimulated with LPS, LPS + Interferon Gamma, Il-4 + IL-13, IL-10 + TGF-Beta, and TGF-Beta for 24 hours. Supernatants were collected, and the release of cytokines measured by Luminex multiplex system (n = ± SE). Heat maps capture the fold induction of each analyte released by engineered lines over the analytes released by parental line in the presence the aforementioned stimulants. White color represented no change red color represented a higher release of cytokines, and blue color demonstrated a decrease in cytokine release in the engineered lines as compared to AHN parental line.

## RNAseq analysis investigates transcriptional similarities and differences between cryopreserved parental and engineered iPSC derived macrophages

We used RNA-seq analysis to investigate transcriptional similarities and differences between parental vs engineered iPSC-derived macrophages to understand differences in transcription profile and uncover underlying mechanisms that might be triggered by perturbations in SNCA, Progranulin, and MeCP2 genes. Transcription tracks were generated from end stage macrophages confirming presence of genetic engineering at transcription level (S9 Fig). The high purity of end stage macrophage cultures (routinely 90–95% pure) confirmed the absence of contaminating cell types to interfere with the analysis. The RNA seq data revealed an upregulation of 431 genes and concomitant downregulation of 276 in SNCA A53T Macrophages, an upregulation of 549 genes and concomitant downregulation of 356 in GRN R493X macrophages, an upregulation of 466 genes and concomitant downregulation of 406 genes in MeCP2HM macrophages when compared to the parental unengineered macrophages (Fig 8). The cumulative analysis of the gene expression profile of all three engineered lines revealed an overlapping set of 70 genes (Fig 8) in common between all three engineered lines in comparison to the parent line. The complete list of all 70 gene are included in S5A–S5C Table in the Supporting Information. The profile of each engineered line is highlighted in a separate table. The overall expression profile indicates a down regulation of 50 genes in the engineered macrophages reflecting impairment of function and an increase in a subset of genes that regulate migration, metabolism and proliferation all key to survival of the engineered lines.

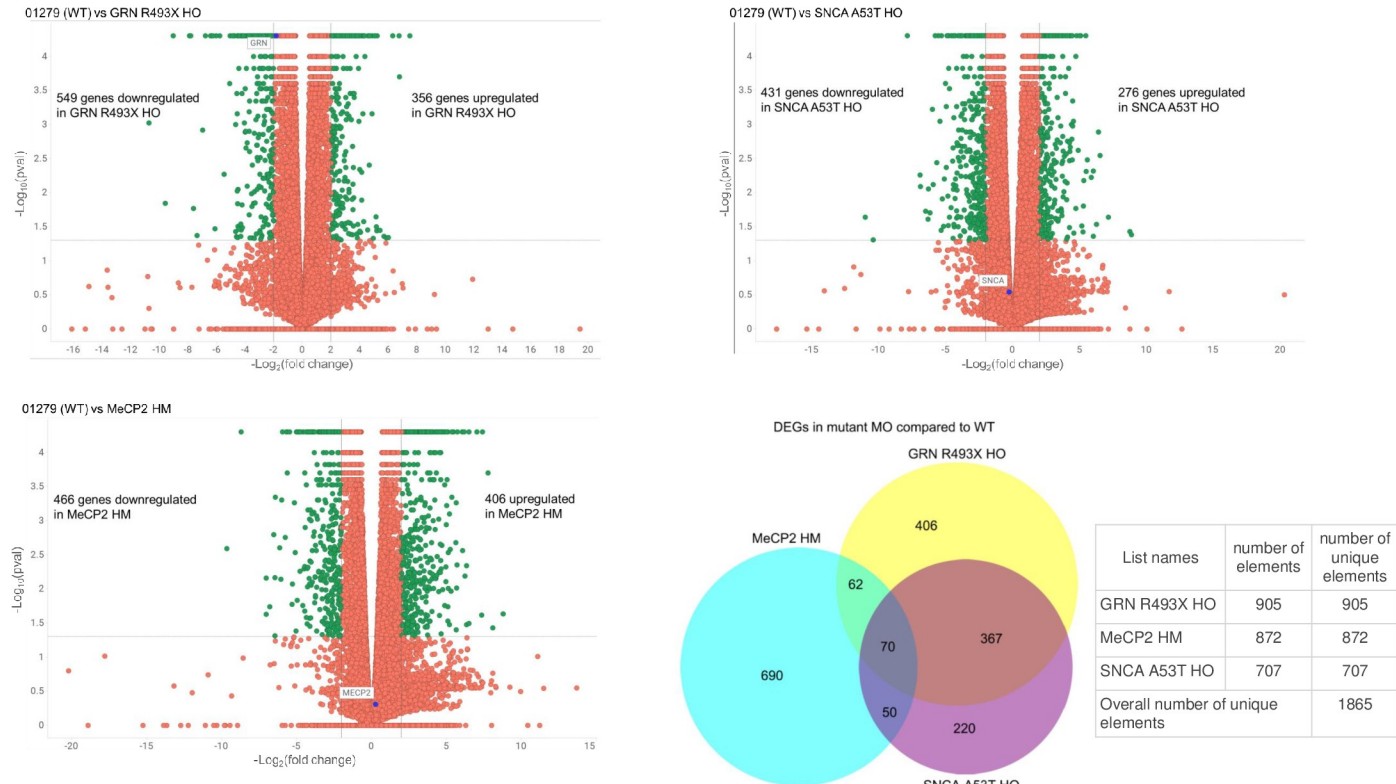

**Fig 8. Differentially Expressed Genes (DEGs) of isogenic iPSC derived macrophages.** Volcano plots of statistically significant differentially expressed genes at corrected P ≤0.05 identified from the RNA-Seq analysis of SNCA A53T macrophage (A), GRN R493X macrophage (B), and MECP2 HM macrophage (C) relative to 01279 parental macrophage. The gene of interest expression is highlighted by a blue circle. Green circles represent statistically significant genes with an absolute fold change ≥ 2. (D) Venn diagram of statistically significant DEGs of isogenic iPSC derived macrophages.

RNA-Seq analysis on the dysregulated genes in engineered macrophages indicated a two-threefold upregulation of CMTM1, ATRNL1, NFKBIZ, MCOLN2, P2RY6, PLEKHG5, GATM and ADGRE4P genes when compared to the unengineered parental macrophages. MTM1 belongs to chemokine-like factor gene superfamily, MCOLN2 gene that plays a role in chemokine secretion, macrophage migration and the regulation of innate immune response and finally ADGRE4P that play a role in adhesion and migration function of macrophages. The upregulation of these genes implied an increased secretion of chemokines facilitating adhesion and migration function in engineered macrophages. The engineered cells also revealed a two-threefold upregulation of ATRNL1 gene that regulates energy homeostasis and GATM gene that affects mitochondrial enzymes supporting the hypothesis for an increase or shift in metabolism post thaw in engineered macrophages. The increase in the levels of NFKBIZ, one of the nuclear I kappa B proteins augments IL-6 production and mis regulation of signaling of the MAPK Erk Pathway implying a bias to the M1 subtype accompanied by an altered cell proliferation kinetics in the engineered lines. An upregulation of ACER2 promotes cell proliferation and survival in engineered lines.

MECP2 HM macrophages revealed a decrease in the expression profiles of FBP1, SPRED1, GATM, ZNF366 COL1A2, TNFSF15, SLC9A9, MPEG1, MARCH1, SEMA3A genes whereas as SNCA A53T macrophages and GRN R493X macrophages displayed and increase in expression of all the above-mentioned genes. A closer look at the gene functions indicated their role in sugar uptake, mitochondrial metabolism, activators of pathways affecting lysosomal and vesicular transport and structural integrity of cells. The downregulation of these genes and the 50 additional genes supports the poor function and analyte profile seen in MECP2 HM macrophages. The gene concept network plots offer a deeper visual analysis to the linkages of genes and biological concepts as a network (Fig 9) to highlight the pathways affected in engineered macrophages regulating cell metabolism, exocytosis, transport, degranulation, immune activation, vascular circulation and neural differentiation pathways.

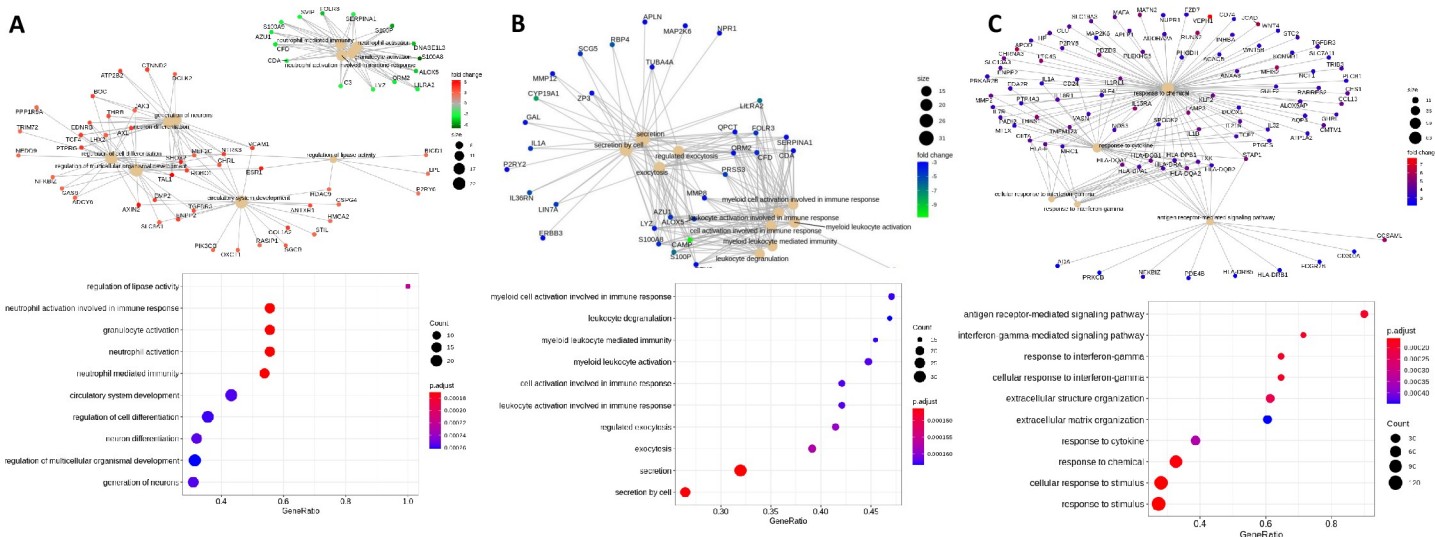

**Fig 9. Functional enrichment analysis visualization.** Gene concept network plots (cnet plots) and dot plots showing the results of GO (biological process) term enrichment analysis performed for all the statistically significant (corrected p-value ≤ 0.05 and |Log2FC| ≥ 2) upregulated and downregulated genes obtained from RNASeq analysis of the SNCA A53T macrophage (A), GRN R493X macrophage (B), and MECP2 HM macrophage (C) relative to 01279 parental macrophage. Dot size indicates *k/n* ratio ("gene ratio"), where *k* is the number of genes participating in the current GO biological process (within the selected gene list), and *n* is the total number of genes annotated as participants of any GO term. Dot color indicates the enrichment test p-value (Fisher's exact test).

## Discussion

Macrophages are crucial orchestrators of both initiation and resolution of immune responses, they play a key role in maintaining homeostasis in most organs of the body. Inflammation is associated with several neurogenerative diseases [4]. Presence of self-maintaining Peripheral Nervous System-resident macrophages that share gene expression patterns with microglia have been reported in aging and neurodegenerative conditions. Farina et al. [53] reported a phenotypic convergence between microglia and peripheral macrophages under pathological conditions suggesting a synergy dictated by the microenvironment. Peripheral macrophages have been reported to act as mediators enhancing the expression of Parkinson's Disease (PD) related genes, such as LRRK2 [9] or show impairment in phagocytosis of amyloid beta in AD patients with ApoE4/4 genotype [7]. Peripheral macrophages can also play a reverse role in calming chronic inflammation and delay the progression of neurodegenerative diseases by reacting to the stimulus like Niacin and polarization to the M2 subtype [8]. This novel information about the biology of microglia and peripheral macrophages sheds new light about their therapeutic potential for neuroinflammatory and neurodegenerative diseases.

The present study describes the development of a defined, scalable differentiation protocol to enable generation of large batches of cryopreserved end stage macrophages from AHN iPSCs as well as isogenically engineered variants that are implicated in various neurological disorders. The method consistently generates a phenotypically pure population of functional end stage macrophages extensively characterized by phenotypic markers. End stage live and cryopreserved macrophages retained viability, purity and phagocytic function. The functional assessment of SNCA A53T, GRN R493X, and MECP2 HM macrophages on eliciting an M1 or M2 dependent responses post stimulation was quantified between live and cryopreserved macrophages from parental and engineered iPSCs. RNA-seq analysis was used to investigate transcriptional similarities and differences between parental and engineered macrophages and the mechanisms that might be underlying the observed differences between parental and engineered macrophages that effect immune responses and function. The results highlight the effect of site-specific genetic alterations in SNCA at A53T and GRN at R493X on the development and temporal relationships between upregulation of genes and secretion of cytokines/chemokines favoring macrophage polarization towards the inflammatory M1 subtype, thus providing new insights into the role of macrophages in neurodegenerative diseases. The data also sheds light on the functional impairment of MECP2 HM macrophages and the subtle differences in functional performance of SNCA A53T and GRN R493X iPSC derived macrophages. Further experimentation is needed to understand the striking changes in metabolism and the interplay between different pathways adapted by macrophages to survive and function post cryopreservation. These findings contribute to understand the effects of perturbation of MeCP2, A53T and GRN at R493X on non-neural lineages. In summary, our study confirms the utility of iPSC-derived macrophages genetically engineered to mimic disease specific mutations a valuable *in vitro* model to study the role of innate immune system in the onset of neurodegenerative diseases.

## Supporting information

**S1 Fig. Sequence confirmation of no off-target mutations in SNCA A53T HO iPSC.** Amplification and sequencing was performed across the TRAK1 (A) or COBL (B) genes covering the two predicted "off-by-one" cut sites for the SNCA A53T nuclease. Sequence confirmation of no off-target mutations in GRN R493X HO iPSC. Amplification and sequencing was performed across the STXBP1 (C) gene at the predicted "off-by-one" cut site for the GRN R493X nuclease.
(TIF)

**S2 Fig. Flowcytometry gating strategy and extended iPSC derived macrophage characterization.** (A) Representative image of the gating strategy used during flowcytometric analysis. The sample is first scatter plat reflects the sample analysis using the FSC vs SSC gates. The live population is a subset of the population gated from the FSC vs. SSC gate stained with Propidium Iodine. Subsequent staining is gated on the live cells to determine single and double positive staining. (B) Parental iPSC and macrophages were stained for the presence of HPC associated markers. The graph denotes the absence of HPC markers on macrophages except CD45. Each value on the graph is an average of ± SE for iPSC and macrophages. (C) Comparative analysis of TREM2 and CD33 expression on iPSC derived microglia and iPSC derived macrophages. Graphs depicts average ± SE for microglia (n = 16) and macrophage (n = 3).
(TIF)

**S3 Fig. iPSC derived macrophage recovery and viability post thaw.** (A) Recovery was quantified by total viable cell number at cryopreservation divided by total viable cell number at thaw. The graphs denote recovery of iPSC derived macrophages post thaw for parental and isogenic disease lines (n = 4 ±SE) and average recovery of parental and engineered macrophages (n = 16 ±SE). (B) Viability of macrophages at harvest (n = 4 ±SE), upon thaw (n = 4 ±SE), and 2 days post thaw (n = 4 ±SE) was quantified by Trypan Blue exclusion on an automated cell counting system.
(TIF)

**S4 Fig. Phagocytotic ability of iPSC derived macrophages from live and cryopreserved cultures.** Phagocytotic function of 01279, SNCA A53T, GRN R493X, and MECP2 HM iPSC derived macrophages was measured by the red fluorescence intensity (RCU) captured on the IncuCyte S3 every 2 hours over the course of 6 days. Total Red Object Integrated Intensity (RCU x $\mu m^2$/image) was analysed using IncuCyte Software (v2019B). The graphs depict the cumulative uptake of pHrodo labeled *S. aureus* bioparticles by live (n = 3 ± SE) (A) and cryopreserved (n = 19 ± SE) (B) macrophages.
(TIF)

**S5 Fig. Secretion of analytes from stimulated and unstimulated isogenic iPSC derived macrophages.** Live and cryopreserved macrophages (three days post thaw) were placed in cytokine free media and stimulated with LPS, LPS + Interferon Gamma, IL-4 + IL-13, IL-10 + TGF-Beta, and TGF-Beta for 24 hours. Release of analytes were quantified from the supernatants using Luminex multiplex system. Each graph depicts an average of triplicate samples ± SE for the specific analyte. The list of the specific analyte are as follows, TNF-alpha (A), IL-6 (B), MMP-9 (C), IL-8/CXCL8 (D), Chitinase 3-like 1 (E), CXCL10 (F), IL-10 (G), CCL2 (H), CD163 (I), IFN-gamma (J), IL-1ra (K), and CCL18 (L).
(TIF)

**S6 Fig. Secretion of analytes from stimulated and unstimulated isogenic iPSC derived macrophages.** Live and cryopreserved macrophages (three days post thaw) were placed in cytokine free media and stimulated with LPS, LPS + Interferon Gamma, IL-4 + IL-13, IL-10 + TGF-Beta, and TGF-Beta for 24 hours. Release of analytes were quantified from the supernatants by Luminex multiplex system. Each graph depicts an average of triplicate samples ± SE for the specific analyte. The list of the specific analyte are as follows, MMP-12 (A), CCL3 (B), CCL22 (C), CCL4(D), IL-1 alpha(E), CCL26 (F), IL-12 p70 (G), IL-1 beta (H), CXCL11 (I), IL-23 (J), CXCL1 (K), and IL-18 (L). Each graph depicts an average of triplicate samples ± SE.
(TIF)

**S7 Fig. Secretion of analytes from stimulated and unstimulated isogenic iPSC derived macrophages.** Live and cryopreserved macrophages (three days post thaw) were placed in cytokine free media and stimulated with LPS, LPS + Interferon Gamma, IL-4 + IL-13, IL-10 + TGF-Beta, and TGF-Beta for 24 hours. Release of analytes were quantified from the supernatants by Luminex multiplex system. Each graph depicts an average of triplicate samples ± SE for the specific analyte. The list of the specific analyte are as follows, PD-L1 (A), CCL1 (B), IL-27 (C), CXCL2 (D), CCL14 (E), CCL20 (F), CCL24 (G), CCL8 (H), CCL13 (I), and Fas Ligand (J). Each graph depicts an average of triplicate samples ± SE.
(TIF)

**S8 Fig. Secretion of analytes from stimulated and unstimulated isogenic iPSC derived macrophages.** Live and cryopreserved macrophages (three days post thaw) were placed in cytokine free media and stimulated with LPS, LPS + Interferon Gamma, IL-4 + IL-13, IL-10 + TGF-Beta, and TGF-Beta for 24 hours. Release of analytes were quantified from the supernatants by Luminex multiplex system. Each graph depicts an average of triplicate samples ± SE for the specific analyte. The list of the specific analyte are as follows, CX3CL1/Fractalkine (A), IL-13 (B), TSLP (C), Granzyme B (D), Alpha-Synuclein (E), TGF-alpha (F), TREM1 (G), CCL17 (H), and CCL11 (I). Each graph depicts an average of triplicate samples ± SE.
(TIF)

**S9 Fig. Visualization of RNASeq reads mapping to the human reference genome.** (A) Representative transcription tracks of 01279 (top panel) and GRN R493X (bottom panel) showing the truncated GRN transcription at amino acid 492. (B) Representative transcription tracks of 01279 (top panel) and SNCA A53T (bottom panel) showing the substitution at amino acid 53. (C) Multiple sequence alignment of assembled transcripts of 01279 and MeCP2 HM showing disrupted mutant transcripts after the puromycin insertion within exon3.
(TIF)

**S1 Table. Oligonucleotides used for gene editing.**
(DOCX)

**S2 Table. Analyses of surface and intracellular pluripotency expression of parental and genetically engineered iPSC lines.**
(DOCX)

**S3 Table. Mean and standard error of Colony Forming Unit (CFU) assay for parental and genetically engineered iPSC lines.**
(DOCX)

**S4 Table. Mean and standard error of macrophage characterization of cell surface antigen expression.**
(DOCX)

**S5 Table.** (a) Statistically significant overlapping differentially expressed genes within SNCA A53T iPSC derived macrophage. (b) Statistically significant overlapping differentially expressed genes within GRN R493X iPSC derived macrophage. (c) Statistically significant overlapping differentially expressed genes within MECP2 HM iPSC derived macrophage.
(DOCX)

**S1 Video. 01279 Live macrophage culture displaying time-lapse imaging phagocytosis of pHrodo labeled *S. aureus* bioparticles.**
(MP4)

**S2 Video. 01279 Cryopreserved macrophage culture displaying time-lapse imaging phago-cytosis of pHrodo labeled *S. aureus* bioparticles.**
(MP4)

**S3 Video. SNCA A53T Live macrophage culture displaying time-lapse imaging phagocyto-sis of pHrodo labeled *S. aureus* bioparticles.**
(MP4)

**S4 Video. SNCA A53T cryopreserved macrophage culture displaying time-lapse imaging phagocytosis of pHrodo labeled *S. aureus* bioparticles.**
(MP4)

**S5 Video. GRN R493X live macrophage culture displaying time-lapse imaging phagocyto-sis of pHrodo labeled *S. aureus* bioparticles.**
(MP4)

**S6 Video. GRN R493X cryopreserved macrophage culture displaying time-lapse imaging phagocytosis of pHrodo labeled *S. aureus* bioparticles.**
(MP4)

**S7 Video. MECP2 HM Live macrophage culture displaying time-lapse imaging phagocyto-sis of pHrodo labeled *S. aureus* bioparticles.**
(MP4)

**S8 Video. MECP2 HM Cryopreserved macrophage culture displaying time-lapse imaging phagocytosis of pHrodo labeled *S. aureus* bioparticles.**
(MP4)

## Author Contributions

**Conceptualization:** Deepika Rajesh.

**Data curation:** Sarah Dickerson, Kiranmayee Bakshy.

**Formal analysis:** Christie Munn, Sarah Burton.

**Investigation:** Christie Munn.

**Methodology:** Christie Munn, Anne Strouse.

**Supervision:** Deepika Rajesh.

**Writing – original draft:** Christie Munn, Deepika Rajesh.

**Writing – review & editing:** Christie Munn, Sarah Burton, Sarah Dickerson, Kiranmayee Bak-shy, Deepika Rajesh.

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
