## [Decision Letter · Decision Letter 0]

9 Oct 2020

PONE-D-20-24606

Generation of cryopreserved macrophages from normal and genetically engineered human pluripotent stem cells for disease modelling

PLOS ONE

Dear Dr. Rajesh,

Thank you for submitting your manuscript to PLOS ONE. After careful consideration, we feel that it has merit but does not fully meet PLOS ONE’s publication criteria as it currently stands. Therefore, we invite you to submit a revised version of the manuscript that addresses the points raised during the review process including:

Detail of the statistical analysis are lackingComparing functional readouts and cell survival of cryopreserved cells directly to cultured cells as benchmark.Add quantitative analysis of cell survival / recovery and function in freeze – thaw experimentThe rationale between establishing a peripheral cell model system (macrophages) and the study of neurological disorders need clarification.Characterize the integrity of potential off-targets that may arise during gene-editing

We ask that the authors address point by point the comments of both reviewers.

We look forward to receiving your revised manuscript.

Kind regards,

Marcel M. Daadi, Ph.D.

Academic Editor

PLOS ONE

Journal Requirements:

"The authors have declared that no competing interests exist. ". 

We note that one or more of the authors are employed by a commercial company: 'Fujifilm Cellular Dynamics,Inc'

Reviewers' comments:

Reviewer's Responses to Questions

**Comments to the Author**

1. Is the manuscript technically sound, and do the data support the conclusions?

Reviewer #1: Partly

Reviewer #2: Partly

2. Has the statistical analysis been performed appropriately and rigorously? 

Reviewer #1: No

Reviewer #2: No

3. Have the authors made all data underlying the findings in their manuscript fully available?

Reviewer #1: No

Reviewer #2: Yes

4. Is the manuscript presented in an intelligible fashion and written in standard English?

Reviewer #1: Yes

Reviewer #2: Yes

5. Review Comments to the Author

Reviewer #1: The authors describe a method for the generation of cryo-preserved macrophages from normal and genetically engineered IPSC. Even though the study has interesting aspects the data provided is insufficient for publication in my view.

Major aspects:

no statistical analysis performed, no information on biological and technical replicates, especially:

- the information how many differentiation's for each iPSC line have been performed to obtain HPC

- how many differentiation's from HPC to Macrophages have been performed

- how big the variation of the reported yields (Fig.3C)was between differentiation's

- source of standard differentiation in Fig 2E

-source of standard differentiation's in the Flow cytometry data originates from (especially Fig.4B+C, as you state in line 377 "macrophages revealed significantly lower expression of CD33,...")

- number of biological technical replicates Figure 5, there also a cell by cell quantification would allow a proper statement, of the amount of positive cells above a certain threshold, and to get a quantitative measure calculation of slopes in the linear sector and reporting of % positive cells at the plateau would be suited

- Cytokine release (Fig.6+7)again number of technical and independent replicates are missing and there is a complete lack of statistical analysis

- furthermore, information on normalization is missing is there some normalization to cell numbers, or was expected that the same seeding density results in the same cell counts.

the authors state that the cells are suited also after cryo-preservation for functional readouts. However, the comparison to directly cultivated cells as benchmark is missing and there is no information on survival post cryo-preservation given

The authors highlight the fact that this is a method describing the generation of cryo-preservable macrophages, but in terms of cryo-preservation the method section is limited to one sentence. In respect that different publications already stated to have trouble and poor recovery after freezing and thawing a detailed freezing and thawing protocol would be essential in this study

As mentioned by the authors in line 435-437 mutations in alpha synuclein are often linked to mitochondrial dysfunction, lysosomal dysfunction and ER stress, and that these phenotypes can be recapitulated by iPSC-derived macrophages, it would be important to proof this statement by experimental data

In general the use of the three disease associated lines in combination with the little detail in the phenotypic characterization, weakens the argumentation of the manuscript.

The Discussion only refers to one publication, in order to set this study into context of previous work the discussion should be largely extended.

The authors should discuss:

- what kind of myelopoieses their protocol aims to resemble

-what can be the reasons in differences of gain

- compare the protocols to other published protocol

Conflict of interest

all authors work for a company commercially selling cells, this should be somehow mentioned

Minor aspects:

Figures:

Fig2A a time scale could be added

Fig2D a quantitative assessment could be added

Fig4A scale bar in A1 different than other scale bars, cell number varies a lot between different genotypes, size seems also different, for a proper morphological assessment zoom in pictures with higher magnification would be essential

Fig4C supplementary information including gating strategy for the markers would increase the understanding

Fig5A wrong labeling (A52T instead of A53T)

Methods:

line 124 "fed using a cocktail of small molecules" either leave it out completely and refer to the previous publication or give the details

line 160 "on matrigel or vitronectin" what is the difference for different coatings in your iPSC maintenance and could it have an impact on the differentiation protocol?

line 188 ""presence of M-SCF and IL-1Beta", did you mean M-CSF?

line 221:"allowing accurate segmentation of the fluorescence images" a detailed setting description of the Incucyte analysis and some representative images of this segmentation in a supplement file would help to judge the data generated

Results:

line 383: "This is the first report presenting and confirming the expression of CD47, CXCR1 on iPSC derived macrophages and the potential application for in vivo engraftment and anti-tumor therapy experiments"

especially the second part is largely overstated

line 390: "The low expression of TREM2 aids to distinguish iPSC derived macrophages from iPSC derived Microglia, which expressing very high levels of TREM2(data not shown)" Since this is a product of fujifilm and you should have it available, an inclusion of the data would be favorable.

Reviewer #2: In the current study, the authors describe integration of disease-related mutations in Parkinson’s disease (PD, SNCA A53T), neuronal ceroid lipofuscinosis (NCL, GRN R493X), and Rett syndrome (MECP2 deletion) into a parental iPSC line (01279), and differentiated into mature macrophages. Cell surface markers were characterized in HPCs, and mature macrophages, and differences in macrophage function and response to inflammatory stimuli were compared. Although the manuscript overall is rather descriptive and does not interrogate functional consequences of various neurological mutations in macrophages, the study yields decent potential as a modelling platform which may facilitate future studies describing dyregulation of innate immune processes in these disorders.

While the intent and general scope of the study is appreciated, several aspects of the manuscript require improvement. As the disease mutations modeled within the studies are neurological disorders, the premise of modeling a peripheral immune cell type is somewhat puzzling. Expression of the targeted loci also require characterization, and issues with experimental rigor need to be improved within the manuscript. This includes statistical analyses and experimental descriptions included in the Figure legends.

Comments:

The most crucial aspect lacking in the study is the characterization of the target proteins at HPC and mature macrophages, at least at the mRNA level. It is essential to determine whether GRN transcripts are indeed downregulated by nonsense-mediated decay, confirm MECP2 deletion and SNCA A53T expression.

Presently, the rationale between establishing a peripheral cell model system (macrophages) and the study of neurological disorders is not clear. If the intention is that the iPSC system could potentially model microglia, more discussion would be required on how the system here can be applied to microglial differentiation. Or if infiltration of macrophages into the CNS may be relevant to disease pathogenesis, this may also be an application relevant to the model used. Without a solid connection between innate peripheral immunity (macrophages) and PD, Rett or CNL, the premise underlying the work as presented remains weak.

Did the authors characterize the integrity of potential off-targets that may arise during gene-editing? Although sequence alterations are shown in Fig. 1A, no apparent analysis on potential non-targeted loci is currently presented.

It is assumed that the mutant lines described were generated as homozygous mutations? It will be important to highlight that mutations described in the study were homozygously generated (for SNCA, GRN). Furthermore, a description of the relevant mutations described in the study in the context of their respective clinical settings will be necessary; for example, SNCA A53T dominantly manifests clinically with a single copy.

With respect to the previous point, it is necessary to provide more description and detail with respect to gender. It appears that 01279 cells are male; further discussion is required with respect to disease manifestation as Rett syndrome manifests primarily in females.

In general, the experimental and statistical descriptions in the Figure legends are far too terse. Number of independent experiments, replicate cultures, and graph descriptions (are graphs mean +/- SE?) have not been described for almost all figures (Fig. 1D, 2E, 4B/C, 5A).

Figure 6 (cytokine release) and 7, are these derived from a single measurement from one clone? If so, this is insufficient to conclusively draw any changes described, for example increased IL-10, IL-12 secretion in GRN R493X macrophages. There do appear to be error bars in Fig. 6A, are these values significant? In fact, there appears to be no statistical analyses whatsoever found in the present manuscript.

The graph in Fig. 6 – why is the baseline set at 1pg/ml? The y-axis origin should be set at “0”.

The heatmaps in Fig. 7 are very problematic, and suggests that every single cytokine evaluated differs from WT 01279 under every condition. No scale is included to depict degrees of change, and use of red to depict smaller changes (even 0) is extremely misleading.

The descriptions in Fig. 5 (phagocytosis) are very poor. No description is seen with respect to the number of replicates, images, and macrophage clones were analyzed. The number of S. aureus particles would also be helpful. It is assumed that graphs depict mean/SE? Is there any statistical analyses for this experiment?

A concluding diagram or table to summarize the findings with respect to the various mutations and their effects on macrophage function would be useful. This would include differences in S. aureus uptake, cytokine release, and response to inflammatory stimuli.

Fig. 3C, the differentiation efficiency is somewhat confusion. Would this not be more clear as an absolute ratio (ratio of 1.0) or percentage? Some confusion arises when the number of macrophages exceeds the seeded iPSCs (1.74). Some explanation may also be warranted why the mutant cell lines yield fewer mature macrophages compared to the parental line.

Fig. 4, percent lethality or recovery would be useful for cells that have been frozen and thawed.

Fig. 2C, the cell surface depiction of HPCs would benefit from a parallel comparison with mature macrophages in the different lines. Also, it would be nice to add the undifferentiated 01279 cell line as a control.

Sequence information would be useful for the donor targeting oligos used during gene editing.

6. PLOS authors have the option to publish the peer review history of their article (what does this mean?). If published, this will include your full peer review and any attached files.

Reviewer #1: No

Reviewer #2: No

---

## [Author Response · Author response to Decision Letter 0]

16 Dec 2020

Enclosed is the summary of all questions addressed by the reviewers and our response to each query: 

1. Detail of the statistical analysis are lacking

Response: All statistical analysis presented in the paper is represented as average number of replicates ± SE. This information has now been included in each figure legend.

2. Comparing functional readouts and cell survival of cryopreserved cells directly to cultured cells as benchmark.

Response: Functional readouts for phagocytosis and cytokine release have been updated to reflect live vs. cryopreserved cells in Figures 5-7.

3. Add quantitative analysis of cell survival / recovery and function in freeze – thaw experiment

Response: Cell survival post thaw has been included in the supplemental section (S3 Fig).

4. The rationale between establishing a peripheral cell model system (macrophages) and the study of neurological disorders need clarification.

Response: Has been addressed in the response to Reviewer’s 2 question below.

5. Characterize the integrity of potential off-targets that may arise during gene-editing

Response: Has been addressed in the response to Reviewer’s 2 question below.

Comments to the Author continued 

Major aspects:

6. Question: no statistical analysis performed, no information on biological and technical replicates, especially: the information how many differentiations for each iPSC line have been performed to obtain HPC

Response: The parental iPSC line has been differentiated to Macrophages at least 32 times. The isogenic engineered iPSC lines have been differentiated to Macrophages in three different independent experimental runs. 

7. Question: How many differentiation's from HPC to Macrophages have been performed-

Response: The manuscript contains the summary of three independent Macrophage differentiation experiments for the parental isogenic lines and the summary of two independent differentiations from the engineered iPSC lines. Since all the differentiation runs originated from the parental and isogenic ally engineered originated from the same hypoxic working cell bank of iPSCs the variation in efficiency was marginal. 

8. Question: source of standard differentiation in Fig 2E

Response: 32 Serum free Methocult dishes were plated with 5000 HPCs derived from parental iPSC or isogenically engineered lines per dish and incubated for 14 days. The emerging colonies were manually were scored to quantify multipotency of HPCs. This information has been included to the legend for Fig2 

9. Question: source of standard differentiation's in the Flow cytometry data originates from (especially Fig.4B+C, as you state in line 377 "macrophages revealed significantly lower expression of CD33,...")

Response: The data represented n Fig 4B and 4C denotes the average of three experimental replicates to quantify CD68 and a panel of Macrophage specific purity markers.

10. Question: The number of biological technical replicates Figure 5, there also a cell by cell quantification would allow a proper statement, of the amount of positive cells above a certain threshold, and to get a quantitative measure calculation of slopes in the linear sector and reporting of % positive cells at the plateau would be suited 

Response: Figure 5 has been updated to reflect the average of three live (n=3) and cryopreserved (n=19) experimental runs to capture the phagocytic function of macrophages. All experiments across all three cell lines were initiated with the same cell number. The comparison between live and cryopreserved Macrophage samples was performed in the same plate using identical reagents. Here is a short write up to explain the quantification of cell viability and function using the IncuCyte system. 

Cell Viability Analysis by Real-Time Live-Cell Imaging System: The IncuCyte® Live-Cell Analysis System (Sartorius Stedim Biotech GmbH, Göttingen, Germany) is an image-based real-time system that allows for an automatic acquisition and analysis of cell images. With the use of two lasers, both phase contrast as well as fluorescence images can be captured. The entire system is placed inside a cell culture incubator in order to guarantee controlled cultivation conditions during real-time monitoring. Phase contrast and fluorescence images are automatically recorded and analyzed using customized software tools in the IncuCyte® S3 image analysis software (Sartorius Stedim Biotech GmbH, Göttingen, Germany). With pre-defined imaging masks, fluorescence signals of the recorded images are then analyzed and counted. Parameters such as minimum fluorescence signal intensity are considered and defined in advance (e.g., to exclude diffuse background noise from the evaluation). The same imaging masks are applied to all acquired images. The data is exported as Counts/Image, which represents the counted fluorescence signals with respect to a single image. The applied dynamic image processing and analysis enables quantitative real-time analyses of fluorescence signals in an imaging field. In addition, by using pre-defined cell-specific imaging masks containing information on cell size and shape, cell growth can be monitored in real-time, by analyzing the occupied area of an imaging field in phase contrast images. Accordingly, this system provides both quantitative and kinetic data.

11. Question: - Cytokine release (Fig.6+7) again number of technical and independent replicates are missing and there is a complete lack of statistical analysis - furthermore, information on normalization is missing is there some normalization to cell numbers, or was expected that the same seeding density results in the same cell counts.

Response: The Luminex data was generated from two independent experiments for each line and each run had a set of triplicate samples for each analyte. The experiment was initiated with the same number of cells with WT and isogenic lines. The data is expressed as fold induction with respect to the WT Macrophages. The fold induction is =1 when the level of analyte is identical to WT Macrophages. 

For generating the benchmark data on the analytes released by stimulated and unstimulated Macrophages, (requested by the reviewers) we stimulated fresh and cryopreserved macrophages under identical conditions. The one big change that happened between the first and the second version of the manuscript submission was the change in the Luminex instrumentation. The earlier set of data was acquired using the Luminex Mag Pix and the data generated in the second run was acquired using the Luminex FlexMAP3D system. The FlexMAP3D system utilizes a different detection method and is more sensitive than the MagPix system.

12. Question: The authors state that the cells are suited also after cryo-preservation for functional readouts. However, the comparison to directly cultivated cells as benchmark is missing and there is no information on survival post cryo-preservation given-

Response: To address this question we initiated a differentiation experiment to generate fresh end stage macrophages from WT and isogenically engineered lines and compared the functional assays that were performed with fresh and cryopreserved macrophages side by side to enable bench marking. Figures 5, 6 and 7 contain a side by side comparison of fresh (noncryopreserved) Vs cryopreserved iPSC derived macrophages.

13. Question: The authors highlight the fact that this is a method describing the generation of cryo-reservable macrophages, but in terms of cryo-preservation the method section is limited to one sentence. In respect that different publications already stated to have trouble and poor recovery after freezing and thawing a detailed freezing and thawing protocol would be essential in this study

Response: A proprietary method to harvest end stage macrophages has been developed by Cellular Dynamics. The end stage cells were harvested and cryopreserved using a control rate freezer with a customized program which further enhanced the survival, recovery and viability of Macrophages post cryopreservation. 

14. Question: As mentioned by the authors in line 435-437 mutations in alpha synuclein are often linked to mitochondrial dysfunction, lysosomal dysfunction and ER stress, and that these phenotypes can be recapitulated by iPSC-derived macrophages, it would be important to proof this statement by experimental data. In general, the use of the three disease associated lines in combination with the little detail in the phenotypic characterization, weakens the argumentation of the manuscript.

Response: We thank the reviewers for highlighting additional functional impairments associated with A53T mutations that been recapitulated in vitro. The molecular analysis of end stage macrophages reveals the presence of the homozygous mutation in end stage macrophages. 

In addition, RNASeq data generated from cryopreserved macrophages highlights the differences in the engineered lines compared to the parental lines which reflects many changes in genes controlling cellular metabolism.

The additional functional assays suggested by the reviewers to demonstrate stress in mitochondria and ER can be included in the future to facilitate mimic disease modelling applications. 

Although the heterozygous mutation is more studied there are reports on the familial and sporadic forms of Parkinson’s disease that contain homozygous A53T mutations. Generating an iPSC line harboring a homozygous A53T mutant alpha-Syn can be used to mimic in vivo neurotoxicity and co culture applications to mimic neurodegeneration.

15. Question: It is assumed that the mutant lines described were generated as homozygous mutations? It will be important to highlight that mutations described in the study were homozygously generated (for SNCA, GRN). 

Response: A description is included in the Material and Methods section stating SNCA A53T and GRN R493X are homozygotes. 

16. Question: Furthermore, a description of the relevant mutations described in the study in the context of their respective clinical settings will be necessary; for example, SNCA A53T dominantly manifests clinically with a single copy.

Response: This has been addressed in the previous question.

17. The Discussion only refers to one publication, in order to set this study into context of previous work the discussion should be largely extended.

Response: Additional papers have been included to the discussion to highlight the role of the engineered mutations for disease modelling applications. These iPSCs could be differentiated to neurons, microglia and astrocytes to set up an elegant organoid system for modelling neurodegenerative diseases.

18. Question: The authors should discuss:

- what kind of myelopoieses their protocol aims to resemble

-what can be the reasons in differences of gain 

- compare the protocols to other published protocol

Response: Hematopoiesis during human development includes the yolk sac- or primitive hematopoiesis and aorta-gonad-mesonephros (AGM) derived deﬁnitive hematopoiesis that gives rise to engraftable hematopoietic stem cells (HSCs). Yolk sac-derived deﬁnitive HSCs along with AGM derived HSCs can generate macrophages. 

The present paper utilizes a feeder free defined and scalable protocol to generate both HPCs and end stage macrophages. The generation of a myeloid progenitor past the HPC stage differs from other published methods on generating myeloid cells from iPSCs which utilize GMCSF (2) or GCSF (5) for myeloid expansion or the purification of monocytes from CMPs to generate end stage macrophages (1)

We think developmentally iPSC derived myeloid cells mimic primitive hematopoiesis. The myeloid progenitors generated the present method retain the potential to generate granulocytes (neutrophils, eosinophils, basophils) and cells of monocyte/ macrophage lineage including DCs and osteoclasts. Their multipotent capability of myeloid cells generated in this protocol mimics the description outlined by Choi et al(1). 

19. Question: Conflict of interest all authors work for a company commercially selling cells, this should be somehow mentioned

Response: We have now stated that all authors are the employees of Fujifilm Cellular Dynamics, Inc; a large-scale manufacturer of human cells, created from induced pluripotent stem cells, for use in basic research, drug discovery and regenerative medicine applications.

Minor aspects:

Figures:

20. Question: Fig2A a time scale could be added 

Response: Fig 2A time scale of HPC differentiation has been included 

21. Question: Fig2D a quantitative assessment could be added 

Response: Fig 2D depicts representative images taken from each cell line. Each line had n=3 slides stained for megakaryocyte staining and a visualization of pink color due to the alkaline phosphatase reaction confirms CD41 expression in megakaryocyte cultures.

22. Question: Fig4A scale bar in A1 different than other scale bars, cell number varies a lot between different genotypes, size seems also different, for a proper morphological assessment zoom in pictures with higher magnification would be essential 

 Response: The scale bar has now been added to the figures to reflect the magnification

23. Question: Fig4C supplementary information including gating strategy for the markers would increase the understanding 

Response: We have included a representative figure in the supplemental section (S2 Fig) to highlight the gating strategy used to capture the staining profile by flowcytometry.

24. Question: Fig5A wrong labeling (A52T instead of A53T) 

Response: The typographic error has been corrected 

Methods:

25. Question: line 124 "fed using a cocktail of small molecules" either leave it out completely and refer to the previous publication or give the details

Response: Small molecule cocktails used are described in Mack et al listed in the bibliography.

26. Question: line 160 "on matrigel or vitronectin" what is the difference for different coatings in your iPSC maintenance and could it have an impact on the differentiation protocol?

Response: AT FCDI iPSCs have been routinely maintained on matrigel or vitronectin without any noticeable differences seen in the downstream differentiation processes as assessed by end stage purity. All iPSC cultures used in this present paper were maintained in the presence of E8 media Matrigel and hypoxic conditions.

27. Question: line 188 ""presence of M-SCF and IL-1Beta", did you mean M-CSF?

Response: We have corrected the typographic error. 

28. Question: line 221:"allowing accurate segmentation of the fluorescence images" a detailed setting description of the Incucyte analysis and some representative images of this segmentation in a supplement file would help to judge the data generated

Response: A more detailed description of the analysis software has been included in the manuscript along with videos of the phagocytosis kinetics of live and cryopreserved macrophages has been included in the supporting information as S1-S8 videos.

Results:

29. Question: line 383: "This is the first report presenting and confirming the expression of CD47, CXCR1 on iPSC derived macrophages and the potential application for in vivo engraftment and anti-tumor therapy experiments"

Response: We have not found any references yet revealing the expression profile of CD47 in iPSC derived macrophages. This is the first paper that validates the expression profile of CD47. The tone of the text has been changed to reflect its usefulness in screening and preclinical applications. 

30. Question: line 390: "The low expression of TREM2 aids to distinguish iPSC derived macrophages from iPSC derived Microglia, which expressing very high levels of TREM2(data not shown)" Since this is a product of Fujifilm and you should have it available, an inclusion of the data would be favorable.

Response: We have included the TREM2 expression profile in iPSC derived Microglia from the same donor line in the supplemental section (S2 Fig). 

Reviewer #2: In the current study, the authors describe integration of disease-related mutations in Parkinson’s disease (PD, SNCA A53T), neuronal ceroid lipofuscinosis (NCL, GRN R493X), and Rett syndrome (MECP2 deletion) into a parental iPSC line (01279) and differentiated into mature macrophages. Cell surface markers were characterized in HPCs, and mature macrophages, and differences in macrophage function and response to inflammatory stimuli were compared. Although the manuscript overall is rather descriptive and does not interrogate functional consequences of various neurological mutations in macrophages, the study yields decent potential as a modelling platform which may facilitate future studies describing dysregulation of innate immune processes in these disorders.

31. Question: While the intent and general scope of the study is appreciated, several aspects of the manuscript require improvement. As the disease mutations modeled within the studies are neurological disorders, the premise of modeling a peripheral immune cell type is somewhat puzzling. Expression of the targeted loci also require characterization, and issues with experimental rigor need to be improved within the manuscript. This includes statistical analyses and experimental descriptions included in the Figure legends.

Response: Inflammation is associated with several neurogenerative diseases (7). Presence of self-maintaining Peripheral Nervous System-resident macrophages that share gene expression patterns with microglia have been reported in aging and neurodegenerative conditions. In a healthy brain, microglia and macrophages are easily distinguished through microscopy by their ramified and amoeboid morphology, and a specific panel of surface and intracellular markers. During neuroinflammation, such as in AD, activated microglia and peripheral macrophages alter their respective morphology and marker expression patterns, confounding their distinction. Many lineage tracing experiments in murine models using labelled or chimeric systems have confirmed the presence of peripheral macrophages in the brain (8). Recently Inlay et al (9) have identified a unique marker CD11a to distinguish PAM and Microglia in murine systems.

Farina et al (10) reported a phenotypic convergence between microglia and peripheral macrophages at certain developmental stages and under pathological conditions. These observations suggest that the two cell types act synergically boosting their mutual activities depending on the microenvironment. This novel information about the biology of microglia and peripheral macrophages sheds new light about their therapeutic potential for neuroinflammatory and neurodegenerative diseases. 

Jairani et al (11) demonstrated that peripheral blood derived macrophages in AD patients with ApoE4/4 genotype were ineffective in phagocytosing Amyloid beta and susceptible to apoptosis. Peripheral macrophages can also act as mediators to enhance the expression of Parkinson’s Disease (PD) related genes, such as LRRK2, by pathogen induced endomembrane damage uncovering a link between membrane damage and onset of Parkinson’s Disease. Presence of M2 macrophages during neural inflammation play a role in calming chronic inflammation and delay the progression of neurodegenerative diseases. Supplementation of PD patients with low dose of Niacin alters the polarization of Macrophages from M1 to M2 suppressing inflammation and progression of PD (12, 13)

Peripheral blood macrophages can also cross BBB, acquire signatures transcripts that are unique to microglia. Resident and infiltrating macrophages contribute to the secretion of pro inflammatory factors and play a role in causing neural damage and play a role in amplification of neurodegeneration. 

The present paper offers a unique tool set of macrophages engineered to express mutations associated typically seen in neuronal cells and study the effect of these mutations in non-neural cell lineages and for gene therapy, co culture pre-clinical disease modeling for neuro degenerative diseases.

Comments:

32. Question: The most crucial aspect lacking in the study is the characterization of the target proteins at HPC and mature macrophages, at least at the mRNA level. It is essential to determine whether GRN transcripts are indeed downregulated by nonsense-mediated decay, confirm MECP2 deletion and SNCA A53T expression.

Presently, the rationale between establishing a peripheral cell model system (macrophages) and the study of neurological disorders is not clear. If the intention is that the iPSC system could potentially model microglia, more discussion would be required on how the system here can be applied to microglial differentiation. Or if infiltration of macrophages into the CNS may be relevant to disease pathogenesis, this may also be an application relevant to the model used. Without a solid connection between innate peripheral immunity (macrophages) and PD, Rett or CNL, the premise underlying the work as presented remains weak.

Response: There are recent papers that highlight the role of macrophages in neurodegenerative diseases as mediators on inflammation and accelerate the progression of the disease. Hence this set of isogenically engineered iPSC derived Macrophages are a one of kind tool kit for disease modelling in vitro along with a blood brain barrier model or DIVI model.

There is a wealth of literature available on the role of microglia in neurodegenerative diseases. We are currently generating a panel microglia with the same engineered lines to provide the end user an elegant system to study the role of both macrophage ad microglia in neurodegenerative disease modelling applications.

33. Question: Did the authors characterize the integrity of potential off-targets that may arise during gene-editing? Although sequence alterations are shown in Fig. 1A, no apparent analysis on potential non-targeted loci is currently presented.

Response: Yes, off-target analysis was performed. A more detailed description has been added to the results section and supporting information (S1 Fig).

34. Question: With respect to the previous point, it is necessary to provide more description and detail with respect to gender. It appears that 01279 cells are male; further discussion is required with respect to disease manifestation as Rett syndrome manifests primarily in females.

Response: Rett syndrome almost exclusively affects females, although males can be affected. These patients have more severe symptoms that develop earlier.

Hemizygotic MECP2 mutations in male lead to fatality and the prevalence of male RTT is extremely rare. This iPSC line offers a unique model to study a complete full null mutant of MeCP2 to understand this fatality and explore the options of downstream implications of this gene in several non-neural and explore options for gene therapy for Rett’s syndrome.

34. Question: In general, the experimental and statistical descriptions in the Figure legends are far too terse. Number of independent experiments, replicate cultures, and graph descriptions (are graphs mean +/- SE?) have not been described for almost all figures (Fig. 1D, 2E, 4B/C, 5A).

Figure 6 (cytokine release) and 7, are these derived from a single measurement from one clone? If so, this is insufficient to conclusively draw any changes described, for example increased IL-10, IL-12 secretion in GRN R493X macrophages. There do appear to be error bars in Fig. 6A, are these values significant? In fact, there appears to be no statistical analyses whatsoever found in the present manuscript.

Response: To address the reviewers queries regarding the differences between fresh and cryopreserved macrophages, a new differentiation was initiated from the same iPSC working cell bank and the end stage fresh cultures were compared to cryopreserved Macrophages side by side under identical experimental conditions. The analytes were quantified, and the data is presented in Figures 6 and 7. The figures depict triplicate samples ± SE values. The samples were run on a newer version of the Luminex, FlexMAP3D with xPONENT 4.3.309.1.

35. Question: The graph in Fig. 6 – why is the baseline set at 1pg/ml? The y-axis origin should be set at “0”. 

Response: The data is now represented in Log Scale and includes the recent data of comparison between fresh and cryopreserved macrophages from parental and engineered iPSC lines.

36. Question: The heatmaps in Fig. 7 are very problematic and suggests that every single cytokine evaluated differs from WT 01279 under every condition. No scale is included to depict degrees of change and use of red to depict smaller changes (even 0) is extremely misleading.

Response: The color scheme has been updated in Fig 7. The scale is denoted in the figure. All experiments were performed at the same time with the same kit /instrument and starting with the same number of input cells and using of the same lot of reagents throughout the experiment.

37. Question: The descriptions in Fig. 5 (phagocytosis) are very poor. No description is seen with respect to the number of replicates, images, and macrophage clones were analyzed. The number of S. aureus particles would also be helpful. It is assumed that graphs depict mean/SE? Is there any statistical analyses for this experiment?

Response: Figure 5 has been updated to reflect live (n=3) and cryopreserved (n=19) macrophage phagocytotic abilities. All wells were seeded at the same cell number. The reported RCU value of Red fluorescent intensity was analyzed using the IncuCyte s3 software with a predefined mask. The graphs depict the calculated RCU values ± SE. Control wells for each cell line were included to eliminate any background fluorescence. A more detailed description has been added to the manuscript and Figure 5 legend. 

38. Question: A concluding diagram or table to summarize the findings with respect to the various mutations and their effects on macrophage function would be useful. This would include differences in S. aureus uptake, cytokine release, and response to inflammatory stimuli.

Response: GO enrichment analysis performed on all engineered lines, figure 9 now depicts upregulated and downregulated genes resulting from perturbations caused by the different mutations.

39. Question: Fig. 3C, the differentiation efficiency is somewhat confusion. Would this not be more clear as an absolute ratio (ratio of 1.0) or percentage? Some confusion arises when the number of macrophages exceeds the seeded iPSCs (1.74). Some explanation may also be warranted why the mutant cell lines yield fewer mature macrophages compared to the parental line.

Response: The total viable cell numbers of input iPSCs required to initiate the differentiation and total cell viable number of output macrophages at the end of the differentiation process are the metrics used to generate the efficiency of the differentiation. The percentage efficiency was calculated by the total viable number of macrophages at the end of differentiation divided by the total viable number of starting iPSCs. Expansion in cell number is observed throughout the differentiation process in the parental line and this expansion in cell number translates to a higher efficiency of the differentiation process in the WT iPSCs compared to the isogenic engineered derived iPSCs.

Our observations are supported by observations made by other investigators, Nissen et al (15) reported that peripheral immune cells taken from PD patients have a lower survival rate when placed in culture. Cronk et al (16) reported on loss in certain populations of macrophages associated with MECP2mutations. Toh et al (17) confirmed the association of Progranulin with cell survival which could be further compromised by mutations to knockout the progranulin. Our results support these conclusions since the efficiency of the engineered lines was lower than that reported by the parental line. 

40. Question: Fig. 4, percent lethality or recovery would be useful for cells that have been frozen and thawed.

Response: The viability post thaw of cryopreserved macrophage cultures has been included in the supporting information (S3 Fig).

41. Question: Fig. 2C, the cell surface depiction of HPCs would benefit from a parallel comparison with mature macrophages in the different lines. Also, it would be nice to add the undifferentiated 01279 cell line as a control.

Response: Staining of HPC markers with undifferentiated 01279 iPSCs has been included in the supporting information (S2 Fig).

42. Question: Sequence information would be useful for the donor targeting oligos used during gene editing.

Response: Sequence information has been provided in the supporting information (S1 Table).

43. Question: PLOS authors have the option to publish the peer review history of their article (what does this mean?). If published, this will include your full peer review and any attached files.

If you choose “no”, your identity will remain anonymous, but your review may still be made public.

no

Do you want your identity to be public for this peer review? For information about this choice, including consent withdrawal, please see our Privacy Policy.

• Reviewer #1: No

• Reviewer #2: No

Bibliography 

1. van Wilgenburg B, Browne C, Vowles J, Cowley SA. Efficient, long term production of monocyte-derived macrophages from human pluripotent stem cells under partly defined and fully-defined conditions. PLoS One. 2013;8(8):e71098.

2. Choi KD, Vodyanik MA, Slukvin, II. Generation of mature human myelomonocytic cells through expansion and differentiation of pluripotent stem cell-derived lin-CD34+CD43+CD45+ progenitors. J Clin Invest. 2009;119(9):2818-29.

3. Shi J, Xue C, Liu W, Zhang H. Differentiation of Human-Induced Pluripotent Stem Cells to Macrophages for Disease Modeling and Functional Genomics. Curr Protoc Stem Cell Biol. 2019;48(1):e74.

4. Cao X, van den Hil FE, Mummery CL, Orlova VV. Generation and Functional Characterization of Monocytes and Macrophages Derived from Human Induced Pluripotent Stem Cells. Curr Protoc Stem Cell Biol. 2020;52(1):e108.

5. Lachmann N, Ackermann M, Frenzel E, Liebhaber S, Brennig S, Happle C, et al. Large-scale hematopoietic differentiation of human induced pluripotent stem cells provides granulocytes or macrophages for cell replacement therapies. Stem Cell Reports. 2015;4(2):282-96.

6. Fabriek BO, Van Haastert ES, Galea I, Polfliet MM, Dopp ED, Van Den Heuvel MM, et al. CD163-positive perivascular macrophages in the human CNS express molecules for antigen recognition and presentation. Glia. 2005;51(4):297-305.

7. Wang PL, Yim AKY, Kim KW, Avey D, Czepielewski RS, Colonna M, et al. Peripheral nerve resident macrophages share tissue-specific programming and features of activated microglia. Nat Commun. 2020;11(1):2552.

8. Yona S, Kim KW, Wolf Y, Mildner A, Varol D, Breker M, et al. Fate mapping reveals origins and dynamics of monocytes and tissue macrophages under homeostasis. Immunity. 2013;38(1):79-91.

9. Shukla AK, McIntyre LL, Marsh SE, Schneider CA, Hoover EM, Walsh CM, et al. CD11a expression distinguishes infiltrating myeloid cells from plaque-associated microglia in Alzheimer's disease. Glia. 2019;67(5):844-56.

10. Grassivaro F, Menon R, Acquaviva M, Ottoboni L, Ruffini F, Bergamaschi A, et al. Convergence between Microglia and Peripheral Macrophages Phenotype during Development and Neuroinflammation. J Neurosci. 2020;40(4):784-95.

11. Jairani PS, Aswathy PM, Krishnan D, Menon RN, Verghese J, Mathuranath PS, et al. Apolipoprotein E Polymorphism and Oxidative Stress in Peripheral Blood-Derived Macrophage-Mediated Amyloid-Beta Phagocytosis in Alzheimer's Disease Patients. Cell Mol Neurobiol. 2019;39(3):355-69.

12. Giri B, Belanger K, Seamon M, Bradley E, Purohit S, Chong R, et al. Niacin Ameliorates Neuro-Inflammation in Parkinson's Disease via GPR109A. Int J Mol Sci. 2019;20(18).

13. Wakade C, Giri B, Malik A, Khodadadi H, Morgan JC, Chong RK, et al. Niacin modulates macrophage polarization in Parkinson's disease. J Neuroimmunol. 2018;320:76-9.

14. Cradick TJ, Ambrosini G, Iseli C, Bucher P, McCaffrey AP. ZFN-site searches genomes for zinc finger nuclease target sites and off-target sites. BMC Bioinformatics. 2011;12:152.

15. Nissen SK, Shrivastava K, Schulte C, Otzen DE, Goldeck D, Berg D, et al. Alterations in Blood Monocyte Functions in Parkinson's Disease. Mov Disord. 2019;34(11):1711-21.

16. Cronk JC, Derecki NC, Ji E, Xu Y, Lampano AE, Smirnov I, et al. Methyl-CpG Binding Protein 2 Regulates Microglia and Macrophage Gene Expression in Response to Inflammatory Stimuli. Immunity. 2015;42(4):679-91.

17. Toh H, Chitramuthu BP, Bennett HP, Bateman A. Structure, function, and mechanism of progranulin; the brain and beyond. J Mol Neurosci. 2011;45(3):538-48.

---

## [Decision Letter · Decision Letter 1]

13 Jan 2021

PONE-D-20-24606R1

Generation of cryopreserved macrophages from normal and genetically engineered human pluripotent stem cells for disease modelling

PLOS ONE

Dear Dr. Rajesh,

Thank you for submitting your manuscript to PLOS ONE. After careful consideration, we feel that it has merit but does not fully meet PLOS ONE’s publication criteria as it currently stands. Therefore, we invite you to submit a revised version of the manuscript that addresses the Minor points raised by the reviewers.

We look forward to receiving your revised manuscript.

Kind regards,

Marcel M. Daadi, Ph.D.

Academic Editor

PLOS ONE

Reviewers' comments:

Reviewer's Responses to Questions

**Comments to the Author**

1. If the authors have adequately addressed your comments raised in a previous round of review and you feel that this manuscript is now acceptable for publication, you may indicate that here to bypass the “Comments to the Author” section, enter your conflict of interest statement in the “Confidential to Editor” section, and submit your "Accept" recommendation.

Reviewer #1: (No Response)

Reviewer #2: (No Response)

2. Is the manuscript technically sound, and do the data support the conclusions?

Reviewer #1: Yes

Reviewer #2: Yes

3. Has the statistical analysis been performed appropriately and rigorously? 

Reviewer #1: Yes

Reviewer #2: Yes

4. Have the authors made all data underlying the findings in their manuscript fully available?

Reviewer #1: No

Reviewer #2: No

5. Is the manuscript presented in an intelligible fashion and written in standard English?

Reviewer #1: Yes

Reviewer #2: Yes

6. Review Comments to the Author

Reviewer #1: The authors addressed all major concerns in their revised version of the manuscript. The quality of the manuscript improved significantly. For full data transparency, please deposit RNAseq data files in a public database (i.e. Geo database) and include the link in the material and method section as well as in the Data availability statement.

Reviewer #2: Efforts towards revising this manuscript are appreciated. A few minor lingering concerns remain, which should be easily edited prior to publication.

It appears that the authors may have missed characterizing expression of key transcripts in the study, namely GRN, MECP2 and SNCA, as commented in the first round of review. Alternatively, the authors can highlight these transcripts in the volcano plots in their transcriptomic analyses in Fig. 8.

The discussion in response to Q31 is appreciated. I would suggest incorporating it in the revised manuscript text.

Q34: There does not appear to be any discussion with respect to the gender of the male 01279 cells. It is important that the audience recognize that male iPSCs have been used. A sentence or two of description would be appreciated (especially in the context of Rett syndrome).

Q36: The color depictions in Fig. 7 are not defined as absolute fold change, or Log fold-change. Is “0” white? If so, it would be advisable to indicate that no change is indicated in white.

RNAseq data should be deposited into a public database (GEO); information regarding the data should be included in Materials/Methods.

7. PLOS authors have the option to publish the peer review history of their article (what does this mean?). If published, this will include your full peer review and any attached files.

Reviewer #1: No

Reviewer #2: **Yes: **Timothy Y Huang

---

## [Author Response · Author response to Decision Letter 1]

9 Mar 2021

All responses to reviewers have been addressed and can be found in the Response to Reviewer's letter.

---

## [Decision Letter · Decision Letter 2]

31 Mar 2021

Generation of cryopreserved macrophages from normal and genetically engineered human pluripotent stem cells for disease modelling

PONE-D-20-24606R2

Dear Dr. Rajesh,

We’re pleased to inform you that your manuscript has been judged scientifically suitable for publication and will be formally accepted for publication once it meets all outstanding technical requirements and addressing Reviewer 2's comments regarding sex as biological variable and figure legend.

Kind regards,

Marcel M. Daadi, Ph.D.

Academic Editor

PLOS ONE

Additional Editor Comments (optional):

Reviewers' comments:

Reviewer's Responses to Questions

**Comments to the Author**

1. If the authors have adequately addressed your comments raised in a previous round of review and you feel that this manuscript is now acceptable for publication, you may indicate that here to bypass the “Comments to the Author” section, enter your conflict of interest statement in the “Confidential to Editor” section, and submit your "Accept" recommendation.

Reviewer #2: All comments have been addressed

2. Is the manuscript technically sound, and do the data support the conclusions?

Reviewer #2: Yes

3. Has the statistical analysis been performed appropriately and rigorously? 

Reviewer #2: Yes

4. Have the authors made all data underlying the findings in their manuscript fully available?

Reviewer #2: Yes

5. Is the manuscript presented in an intelligible fashion and written in standard English?

Reviewer #2: Yes

6. Review Comments to the Author

Reviewer #2: Efforts to revise the manuscript are appreciated. All concerns have been satisfied with the following exceptions. Although I have recommended acceptance and publication, it would be appreciated if these issues were addressed in a final revision.

1) Point 3. “There does not appear to be any discussion with respect to the gender of the male 01279 cells. It is important that the audience recognize that male iPSCs have been used. A sentence or two of description would be appreciated (especially in the context of Rett syndrome).” Am I reading correctly that the authors have deleted the following:

“Though Rett Syndrome predominantly impacts females, males can be affected. The onset of

symptoms develops earlier and are often more severe [23, 24].”

I would have rather liked the authors to expand on this (frequency of males to females) rather than to delete this point.

2) There appears to be no figure legends for the supplemental figures, so I am hoping I am interpreting. Figure S9 correctly. While the inclusion of the transcription tracks for GRN and SNCA are appreciated, characterization of MECP in MECP KO cells would also be helpful. It appears that the genomic sequence characterization for MECP KO cells are shown.

7. PLOS authors have the option to publish the peer review history of their article (what does this mean?). If published, this will include your full peer review and any attached files.

Reviewer #2: No

---

## [Editor Report · Acceptance letter]

7 Apr 2021

PONE-D-20-24606R2 

Generation of cryopreserved macrophages from normal and genetically engineered human pluripotent stem cells for disease modelling 

Dear Dr. Rajesh:

I'm pleased to inform you that your manuscript has been deemed suitable for publication in PLOS ONE. Congratulations! Your manuscript is now with our production department. 

Kind regards, 

on behalf of

Dr. Marcel M. Daadi 

Academic Editor

PLOS ONE